# Social threat avoidance depends on action-outcome predictability
Matteo Sequestro [1] ✉, Jade Serfaty[1], Julie Grèzes [1,3] ✉ & Rocco Mennella [1,2,3]

Avoiding threatening individuals is pivotal for adaptation to our social environment. Yet, it remains unclear whether social threat avoidance is subtended by goal-directed processes, in addition to stimulus-response associations. To test this, we manipulated outcome predictability during spontaneous approach/avoidance decisions from avatars displaying angry facial expressions. Across three virtual reality experiments, we showed that participants avoided more often when they could predict the outcome of their actions, indicating goal-directed processes. However, above-chance avoidance rate when facing unpredictable outcomes suggested that stimulus-response associations also played a role. We identified two latent classes of participants: the "goal-directed class" showed above-chance avoidance only in the predictable condition, while the "stimulus-response class" showed no credible difference between conditions but had a higher overall avoidance rate. The goal-directed class exhibited greater cardiac deceleration in the predictable condition, associated with better value integration in decision-making. Computationally, this class had an increased drift-rate in the predictable condition, reflecting increased value estimation of threat avoidance. In contrast, the stimulus-response class showed higher responsiveness to threat, indicated by increased drift-rate for avoidance and increased muscular activity at response time. These results support the central role of goal-directed processes in social threat avoidance and reveal its physiological and computational correlates.

Facial emotional displays convey critical information about the sender's affective state and associated behavioral intentions, but also about potential threats and/or opportunities in the environment[1]. Such information helps us to shape adaptive social behavior, such as avoiding threatening individuals and approaching friendly ones[2,3]. Although there is a consensus that emotions are intimately linked to action, the nature of such a link is still a matter of debate[1,4].

Responses to socio-emotional stimuli have been thought to be strongly influenced by simple motor tendencies to approach or avoid, selected by evolution and/or development for their adaptive value[5–8]. Such action tendencies are determined by the association between the representation of a specific stimulus and the representation of a specific action[9], and can either be innate or formed by associative learning (i.e., by habit formation[7]). In the case of angry facial displays, individuals may learn throughout life that such emotional expressions predict the risk of an aggressive confrontation by personal or vicarious experience[10,11]. In this sense, habit formation may consolidate stimulus-response (SR) associations thought to be automatically

activated when seeing threatening faces, for example, to facilitate fast reactions to threat. Many emotional accounts assume SR associations to be the default determinants of behavior in response to social threat, while cognitively costly and non-automatic goal-directed (GD) processes would only control and refine automatic tendencies[8,12,13]. This regulatory process is goal-directed as it is oriented toward one or more goals that may or may not be congruent with the initial SR tendency[7]. This architecture is considered efficient in selecting responses to threat, as automatic SR tendencies allow swift action selection in response to danger, while slower GD processes enhance decision optimality[14].

More recently, however, the dichotomy between SR and GD processes has been criticized as over simplistic[9,15,16]. Specifically in relation to defensive behavior, SR and GD processes are nowadays conceived on a continuum, where SR processes, leading to both habitual and innate defensive responses (e.g., fight/flight), prevail under imminent threat, whereas GD processes dominate during safety and pre-encounter states (i.e., before the threat is detected in the environment[17–19]). Importantly, recent research in

[1]Cognitive and Computational Neuroscience Laboratory (LNC 2), Inserm U960, Department of Cognitive Studies, École Normale Supérieure, PSL University, 29 rue d'Ulm, 75005 Paris, France. [2]Laboratory of the Interactions between Cognition Action and Emotion (LICAÉ, EA2931), UFR STAPS, Université Paris Nanterre, 200 avenue de La République, 92001 Nanterre, Cedex, France. [3]These authors jointly supervised this work: Julie Grèzes and Rocco Mennella. ✉e-mail: matteo.sequestro@gmail.com; julie.grezes@ens.psl.eu

humans[20–24] and rodents[25–29] converge in assigning a more important role to GD defensive responses than previously thought, both under low and relatively imminent threat[7,9,30]. Indeed, not all GD processes are slow and costly[31], as complex information integration can lead to fast, implicit, and even unconscious estimates of the best action-outcome option, combining automaticity with optimality[7,32–34]. In line with this, when threat is present but not yet attacking and a fast decision needs to be taken (i.e., post-encounter phase), a mixture of "cognitive" (GD) and "reactive" (SR) defensive modes emerges[18,35,36], raising the question on how the brain arbitrates between the two. Notably, social threats, such as encounters with angry or aggressive individuals, most likely lie in this transition zone.

To test whether GD or SR processes, or a combination of both, are involved in determining behavior, a classical approach consists in manipulating the outcome value of a response (e.g., by outcome devaluation[37]) or the action-outcome contingency (e.g., by contingency degradation[37]). Indeed, while GD behavior is sensitive to outcome values and action-outcome contingencies, SR behavior is not affected by learning or environmental feedback[37,38]. It has been proposed that the controllability of expected action-outcomes governs the balance between SR and GD processes[39]. Specifically, when an action reliably leads to a predictable consequence (controllable action-outcome context), GD processes should be privileged as they are more optimal. Conversely, in uncontrollable contexts, when the contingency between actions and consequences is stochastic, the additional computational complexity of GD processes is unnecessary, and SR associations are a more parsimonious policy[39,40].

The current study aimed at characterizing the contribution of SR and GD processes to spontaneous avoidance of threatening avatars in Virtual Reality (VR). Participants had to choose to enter one of two elevators, each containing a virtual avatar that displayed an angry or a neutral facial expression. We manipulated the controllability of action-outcomes, by comparing two conditions in which threat-related action-outcome associations were predictable and unpredictable, respectively. For example, choosing to enter the elevator with the neutral avatar in the predictable condition deterministically resulted in the predicted outcome (i.e., avoidance of the angry avatar), guaranteeing full controllability. Conversely, choosing to enter the elevator with the neutral avatar in the unpredictable condition could stochastically result either in anger avoidance or approach, precluding controllability of the outcome. Importantly, participants were instructed to make spontaneous decisions, and the threatening expressions of the avatars were never explicitly mentioned.

In the predictable condition, we expected participants to avoid the angry avatar more than chance, regardless of whether avoidance is supported by GD processes, SR processes, or a combination of the two. In the unpredictable condition, if GD processes are involved in the decision, avoidance rates should decrease, because of the stochasticity in action-outcome contingencies. Conversely, if avoidance is solely driven by SR associations, no reduction in avoidance rates should be observed. Importantly, a simple reduction in avoidance rate under unpredictability would not rule out the involvement of SR processes in the decision, since SR processes might just keep operating in the unpredictable condition as they do in the predictable one. However, an avoidance rate at chance level in this condition would suggest that SR processes are not driving avoidance, either because they are not elicited in response to threatening expressions, or because they are not overtly influencing avoidance. Therefore, by comparing the avoidance patterns between the two conditions, we tested the effect of GD and SR processes over social-threat avoidance.

## Methods
The presented studies were not preregistered.

### Participants
We conducted three experiments and for each experiment we ran power simulations[41] to determine the sample size required for detecting an effect of the Condition (Predictable vs. Unpredictable) on avoidance choices with 80% power, each time based on the results of the previous experiment (for

Experiment 1, see Supplementary Note 1 for a detailed presentation of the pilot experiment on which we based the simulation). The simulation approach for power calculation involves fitting a Generalized Linear Mixed Model (GLMM) to an initial dataset to extract fixed and random effect parameters. These parameters are then used to generate data for a specified number of artificial participants, on which the same GLMM is fitted. This process is repeated 1000 times for each sample size to test. The power is calculated as the proportion of artificial datasets that show a significant effect ($p < 0.05$) of the desired parameter out of the total 1000 datasets[41]. Compared to analytical methods for power calculation, the simulation approach is suitable for GLMMs and accounts for inter-individual variability (i.e., the random structure of the model). Participants in all experiments were aged between 18 and 35 years old, had no history of neurological or psychiatric disorders (a full list of inclusion criteria is provided in Supplementary Methods 1) and were recruited using convenience sampling through flyers on the university campus and online advertisements.

*Experiment 1:* The power simulation indicated a required sample of 20 participants. Since in the pilot experiment we had data losses due to cybersickness and since the task design was modified to reduce this risk, we decided to aim for 60 participants. Sixty-five volunteers participated in this experiment. We excluded data from 3 participants due to technical issues, 1 due to cybersickness, and 1 because they reported already being familiar with the task. The final sample consisted of 60 participants (*n* = 30 females, *n* = 30 males; mean age ± SD = 23.51 ± 3.83). Data for this experiment was collected between July and October 2022.

*Experiment 2:* The power simulation indicated a required sample of 30 participants. Thirty-two volunteers participated in this experiment. We excluded data from 1 participant due to cybersickness and 1 because they showed lack of understanding of the task instructions. The final sample consisted of 30 participants (*n* = 15 females, *n* = 15 males; mean age ± SD = 22.60 ± 3.29). Data for this experiment was collected between December 2022 and January 2023.

*Experiment 3:* The power simulation indicated a required sample of 60 participants. Sixty-six volunteers participated in this experiment. We excluded data from 3 participants due to cybersickness and 3 due to technical issues. The final sample consisted of 60 participants (*n* = 32 females, *n* = 28 males; mean age ± SD = 23.42 ± 4.59). Data for this experiment was collected between April and June 2023.

For all experiments sex and age were self-reported by participants and no data about race/ethnicity was collected as it is illegal in France, except under special circumstances. The experimental protocol was approved by INSERM, licensed by the local research ethics committee (IRB00003888 – Avis 18-544-ter – 25.10.2021). Participants provided written informed consent and were compensated for their participation. See Supplementary Table S47 for a detailed description of the three samples.

### Materials (experiments 1, 2 and 3)
**Equipment and software.** A virtual environment was presented through the Virtual Reality Head Mounted Display (VR HMD) Oculus Quest 2 (Meta Quest). The Oculus HMD has a resolution of 1920*1832 (per eye). Two Oculus Touch Controllers (one per hand) were used to collect participant responses. The VE was developed using the Unity game engine (version 2020.3.19f1, Unity Technologies) and the UXF package (Unity Experiment Framework[42]). The vertical Field of View of the Oculus device was set to 60 degrees (per eye), and the virtual environment was updated at a constant frequency of 60 Hz.

The software was run on an Alienware Aurora Ryzen Edition PC with a 3.00 GHz 12-core AMD Ryzen 9 5900 processor, a 32GB DDR4 3400 MHz RAM, and an NVIDIA GeForce RTX 3080 10GB GDDR6X graphic card.

**Stimuli.** The virtual environment contained dyads of full-body virtual avatars presenting neutral and angry face expressions selected from the Radboud Face Database[43]. The faces of each dyad of avatars to be presented together were matched for trustworthiness and threat traits[44]. The 2D faces were converted into 3D head models and merged into 3D body

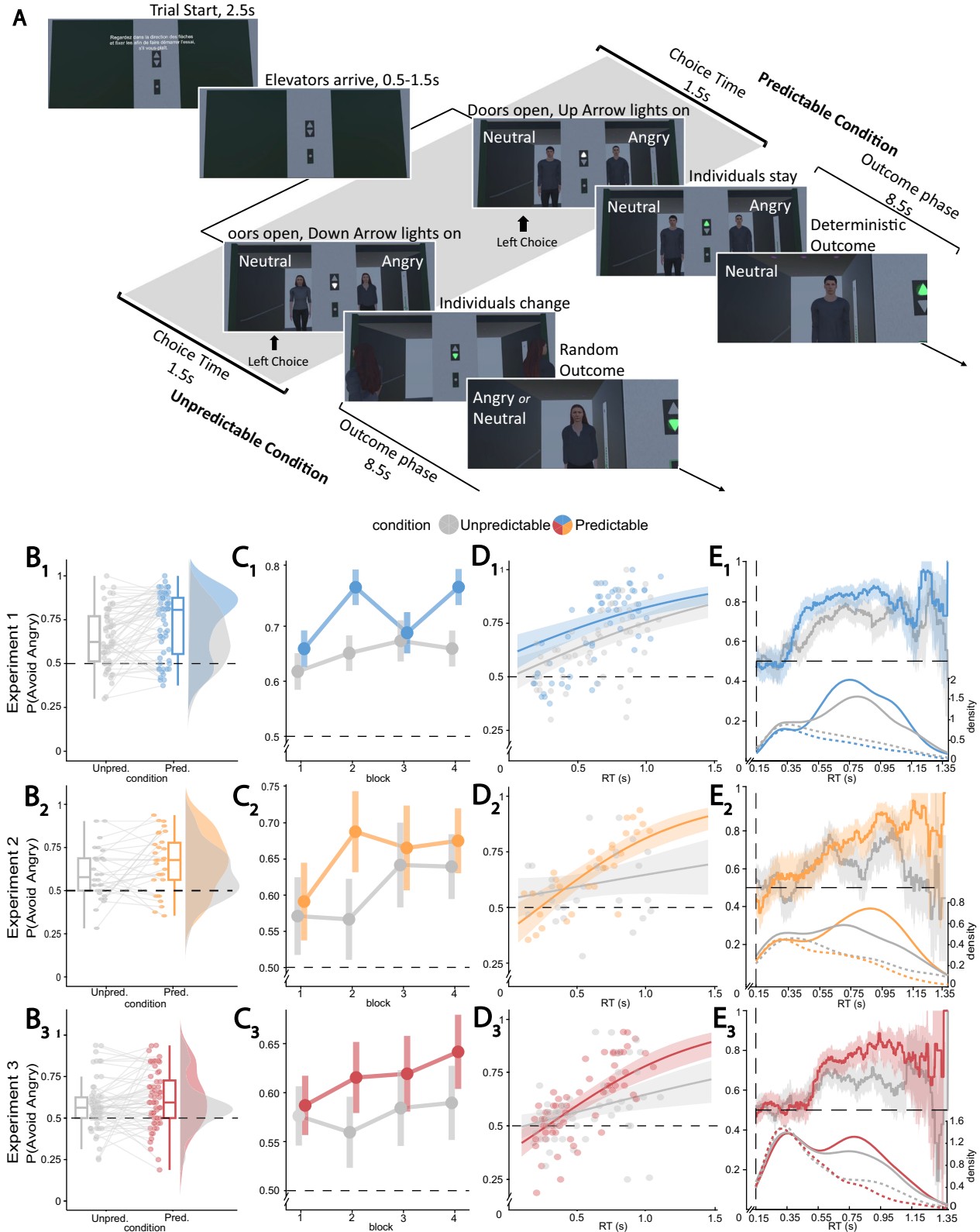

models. We selected images for 10 individuals for each sex, resulting in 10 individuals × 2 sexes (male and female) × 2 facial expressions (neutral and angry), for a total of 40 avatars. Later, we discarded 4 avatars (a dyad for each sex) due to their poor quality in virtual reality, leaving us with a total of 36 stimuli. Detailed information about the creation of the stimuli used in this study can be found in Supplementary Methods 2.

**VR Task**. In the virtual environment, participants faced a wall with two elevators and a frame with two arrows (one pointing up and one pointing down) between them (Fig. 1A). Each trial began as soon as participants fixated the central arrows for 2.5 s. At the elevators' arrival the doors opened, revealing a 3D avatar inside each elevator. Two-thirds of the trials presented an avatar with an angry face and an avatar with a neutral

**Fig. 1 | Task structure and avoidance rates. A** In the Elevator Task, a trial began with a screen that required fixation on the central arrows (2.5 s). After a jittered interval, the two doors opened and simultaneously one of the two arrows lit, indicating whether the elevators were going up or down (anticipating avatars' movements after the choice). Participants had 1.5 s to freely choose which elevator to enter (by pressing the left or right button). After the choice, in the Predictable condition (up arrow lit), participants were led next to the avatar in the chosen elevator during a period lasting 8.5 s (4 s moving inside the elevator, 4.5 s standing next to the avatar). In the Unpredictable condition (down arrow lit), after participants made their choice, the avatars in the elevators walked outside the environment, while two new different avatars (not previously visible) walked inside the elevators. Thus, participants were led next to a new avatar (50% probability of being angry or neutral). At the end of the trial, participants were "teleported" back to the starting position. The scene reset between two trials within 0.5 s. Note that the "Angry" and "Neutral" texts are for illustrative purposes only and were not present in the virtual environment. The picture was created as an illustrative example of the task: avatars' faces were modeled from the Radboud face database[43], and the 3D body models were purchased from the RenderPeople website (https://renderpeople.com). Both sources permit the use of stimuli as examples. **B** Proportion of avoidance choices, P(Avoid Angry), by Condition (gray: Unpredictable; colored: Predictable). Box-plots represent first, second (median) and third quartiles. Whiskers are drawn within the 1.5 interquartile range. Points represent individual participants. **C** Proportion of avoidance responses by Condition (Unpredictable vs. Predictable) and experimental block. Vertical lines: within subjects 95% confidence interval. **D** Predicted probability of avoidance choices as a function of RTs (solid lines). Shaded areas: 95% credible intervals. Points: observed avoidance rates over median RTs for each participant. **E** Observed proportion of avoidance choices as a function of RTs within a 100 ms sliding window. Shaded areas: 95% bootstrapped Confidence Intervals. On the bottom: defective density distribution (i.e. weighted by the relative frequency of responses) of RTs for avoidance (solid) and approach (dotted) responses. In all plots, the black dashed line represents the chance level (50%). **B–E - 1**: Experiment 1 (predictable condition in blue, n = 60), **2**: Experiment 2 (predictable in orange, n = 30), **3**: Experiment 3 (predictable in red, n = 60). Unpred Unpredictable condition, Pred Predictable condition, RT Response Time.

face (*Threat* trials), while one-third of the trials presented the two avatars with only neutral faces (*Neutral* trials). While most analyses were focused on participants' behavior in Threat trials, the addition of Neutral trials was motivated by the goal of dissimulating as much as possible the presence of threatening expression. Furthermore, this allowed us to run control analyses by testing differences between Threat and Neutral trials in terms of response times and physiological responses (Experiment 3).

Participants were asked to freely choose which elevator to enter as quickly as possible after the doors were fully opened. Notably, in Threat trials, participants could choose to enter the elevator with the angry avatar (approach response) or the one with the neutral avatar (avoidance response). The avatars were presented in the periphery of the participants' visual field at a visual angle of 20°. Participants were instructed to maintain fixation on the central arrows during choice. They held in each hand a VR response controller to provide left/right responses and had limited time to make a choice (i.e., 1.5 s). If the response was not given in time, the trial ended and a red text prompt "*Plus rapide!*" ("*Faster!*") was displayed in the virtual environment.

Importantly, the arrow at the center of the scene served as the cue associated with the Condition (Predictable or Unpredictable). Participants were informed that when the arrow pointing up lit (Predictable condition), the avatars would remain in their respective elevators after choice. For example, when participants clicked on the left button, an automatic camera movement after the choice led them next to the avatar present in the left elevator at the moment of choice. Therefore, participants could predict, and thus control in a deterministic fashion, the avatar to be next to (or far from). Conversely, participants were informed that when the down arrow lit (Unpredictable condition), the avatars in both elevators would walk outside after the choice and leave the virtual environment, while two new avatars (not previously visible) would enter the elevators. Importantly, the face identities of the new avatars were always different from those of the avatars present at the moment of choice. Participants were unaware that the new avatars could display an angry or a neutral expression with a 50% probability. Therefore, they had no control over whether they would ultimately approach or avoid angry avatars in Threat trials. Participants were led next to the avatar in the chosen elevator only after the new avatar entered. At the end of each trial, participants were "teleported" back to the starting position and the scene reset within 0.5 s between trials.

It must be stressed that participants were only instructed to freely choose to enter the elevator they preferred. They were further informed of the contingency between the cue (the arrow between the elevators) and the fact that the avatars would stay or leave the elevators after participants' response. No other explicit instructions were given, and the presence of threatening expressions was never mentioned. Overall, we chose this experimental setting to investigate spontaneous avoidance behavior. Furthermore, note that the terms "Predictable" and "Unpredictable" do not refer to the avatars' movements, which were known and explicit, but

to the outcome of the choice in terms of anger approach/avoidance in Threat trials.

The task consisted of one to three practice blocks of 12 trials each, followed by four main blocks of 24 trials each. Between each block, participants were given a 1.5-minute break during which they could remove the VR HMD. To prevent cybersickness, these breaks could be extended for as long as participants wished. All participants completed a first practice block to familiarize with the virtual environment and the task and completed one or two additional practice blocks if their response accuracy was below 80% (if responses were not made within 1.5 s). After each block, participants received feedback indicating the percentage of responses they had provided in the correct timing. If this percentage was lower than 80%, they were asked to make faster decisions.

Avatars with angry expressions were presented only in the main blocks and not in the practice blocks. In each block, half of the trials were from the Unpredictable condition and half from the Predictable condition. Each trial presented only avatars of the same sex, including the new avatars that entered the scene after the choice in the Unpredictable condition. The side of the elevator with the angry avatar (left, right) and the sex of the avatars in the elevators were randomized and counterbalanced throughout the task.

Finally, there was a slight modification of the task between Experiment 1 and Experiments 2 and 3. In Experiment 1, immediately after trial initiation, one of the two arrows lit signaling the arrival of the elevators and, after a jittered time interval (mean interval ± SD = 0.9 s ± 0.3), the doors opened revealing a 3D avatar inside each elevator (Supplementary Fig. S12). In Experiments 2 and 3, the arrow lit simultaneously with the opening of the doors (Fig. 1A; videos of both versions of the task are provided in the OSF repository associated with this study[45]). To prevent participants from anticipating responses, we maintained the jittered interval but shifted it before the onset of the stimulus. The duration of the interval remained the same as in Experiment 1 (mean interval ± SD = 0.9 s ± 0.3).

This choice was motivated by results from Experiment 1, which suggested that participants initiated their decision at the onset of the cue, rather than at door opening. Accordingly, the trial-by-trial duration of this jittered interval in Experiment 1 was negatively correlated with Response Times (RTs; Gamma GLMM with identity link function: β = −0.048; p = 0.005). We reasoned that this was problematic, as a longer time to decide may favor GD processes, by facilitating implementation intentions (i.e., goal-directed plans that automatically link anticipated situational cues to behavioral responses[46–48]). Furthermore, since RTs recorded from the moment the doors opened may not capture the entirety of the decision process, this could bias models that include response times, such as the drift-diffusion models (see below).

**Self-Report Measures.** Numerous studies have shown that social interactions and decision-making can be affected by changes in positive affect[49], depression[50], anxiety and social anxiety[51], as well as reward/

punishment sensitivity[52]. Therefore, participants completed a series of self-administered questionnaires to account for these variables in our sample. Of note, characterizing the effect of these affective and personality traits on behavior is beyond the scope of the present article, which focuses on the balance between SR and GD processes in social threat avoidance. We nonetheless include descriptive results from self-report measures including the observed Cronbach alphas in Supplementary Table S47.

The administered questionnaires were:

- The Positive and Negative Affect Schedule (PANAS[53]), to measure affect;
- The Patient Health Questionnaire depression scale (PHQ-8[54]) to measure depressive symptoms;
- the State Trait Anxiety Scales (STAI[55]) and the Liebowitz Social Anxiety Scale (LSAS[56]), to measure anxiety and social anxiety;
- The Behavioral Inhibition/Behavioral Activation scales (BIS-BAS[57]).

Finally, to track possible cybersickness symptoms due to VR, participants filled in the Simulator Sickness Questionnaire[58].

## Materials specific to Experiment 3: physiological and subjective measures

Previous studies have shown that phasic variations in heart rate (HR) are associated with distinct stages of approach and avoidance decisions. Specifically, the magnitude of cardiac deceleration around the presentation of a threatening stimulus has been associated with the efficiency of information sampling and action preparation, while subsequent cardiac acceleration has been associated with action execution[59–61]. Rather than being a passive reaction to threatening situations, bradycardia has also been considered as an active state that facilitates the preparation of value-based actions and the selection between different action possibilities[60,62]. To test for the association between cardiac deceleration and value integration during the decision, we recorded ECG during the main task, and subsequently ran a subjective evaluation task, asking for an explicit subjective evaluation of each possible outcome (approach/avoidance) encountered in the main task. We predicted that subjects would subjectively prefer avoidance vs. approach scenarios and that heart rate deceleration would be more pronounced under threat in predictable contexts, facilitating the integration of subjective values into the decision[60].

Also, as an exploratory analysis, in Experiment 3, we measured EMG activity of response effector muscles during the main task. Muscle activity, such as involuntary hand muscles contraction following transcranial magnetic stimulation of the primary motor cortex, measured with EMG[63,64], or sustained force exerted due to voluntary isometric wrist extension[65], has been used as an objective marker of action readiness during viewing of emotional images. Similarly, we measured muscle contraction of the *flexor pollicis brevis* with EMG, at the moment of response production, as a measure of action readiness. Here, we wanted to explore whether action readiness is modulated either by the simple presence of threat in the scene, or by the predictability of the outcome, which would clarify whether both SR and GD mechanisms can influence motor preparation.

All physiological data were collected using ADInstruments hardware (ADInstrument, Amsterdam, NL) and amplified prior to digitalization (Dual BioAmp, ADInstrument). All the processing steps and analyses were conducted using custom scripts on MATLAB[66] (R2022b).

### Electrocardiography (ECG) recording and preprocessing.
ECG data were recorded continuously at a sampling rate of 1KHz using three disposable adhesive Ag/AgCl electrodes placed under the left collar bone (ground electrode), right collar bone (negative electrode), and below the left ribs (positive electrode). Raw data were band-pass filtered between 1 and 100 Hz, and R peaks were automatically detected using a custom amplitude-threshold method individually calibrated for each participant. The detected peaks were visually inspected to check for errors in the algorithm and manually corrected when necessary. Inter-Beat-Intervals

(IBIs) were computed as the time difference in milliseconds between two consecutive R-peaks. IBIs that differed in value more than 30% from the previous one were considered artifacts and replaced with a cubic spline interpolated value. Finally, IBIs were converted to instantaneous Heart Rate (iHR) by dividing each of them by 60,000.

To extract event-related cardiac responses, we constructed stimulus-locked iHR time series in each trial by linear interpolation to achieve a sampling rate of 10Hz[67]. The time series thus obtained were band-pass filtered between 0.01 and 2 Hz with a second-order zero-phase forward and reverse Butterworth filter to remove slow drifts and smooth the angles introduced by the interpolation[67]. Finally, the resulting iHR series were epoched from −4 to 10 s around the stimulus onset and baseline corrected by subtracting the mean value of a segment from −2 to −1 seconds before stimulus onset. Note that we did not use a baseline right before the stimulus onset to avoid correcting for a time segment already active due to anticipatory cardiac deceleration. Data from 5 participants was excluded from ECG analysis due to poor signal quality (remaining sample: $N = 55$, $n = 30$ females, $n = 25$ males; mean age ± SD = 23.25 ± 4.58).

### Electromyography (EMG) recording and processing.
EMG activity of the *flexor pollicis brevis* muscles was recorded continuously at 2 kHz using two 4 mm circular Ag-AgCl shielded electrodes attached to each hand as recommended[68]. Prior to electrode fixation, the skin was lightly abraded with an exfoliant gel (Nuprep - Weaver and co.) and then wiped with a damp cloth. Two electrodes per hand were placed in a bipolar montage parallel to the muscle fibers, along an axis connecting the ulnar aspect of the metacarpophalangeal joint and the pisiform, with one electrode placed midway along this axis and the other electrode placed 2 cm (center to center) away from the first electrode[69]. A ground electrode was placed on the *processus spinosus* of the C7 vertebra[70].

The EMG signal was band-pass filtered between 20 and 500 Hz with a zero-phase forward and reverse 4th order Butterworth filter, while spectral interpolation was used to remove power line noise[68]. The signal was then full-wave rectified and EMG envelopes were created by low-pass filtering the signal at 10 Hz with a zero-phase forward and reverse 4th order Butterworth filter. We used a single-threshold algorithm[71] to detect onsets and offsets of EMG bursts. We identified a baseline as the segment with the lowest amplitude in a moving average with a window of 500 msec from 1 s before the stimulus onset to the response time[72]. We defined EMG bursts as the series of contiguous sample points 3 baseline standard deviations above the baseline mean, and the RT generating burst as the EMG burst containing the response time. We defined burst onset and offset as the first and last samples above this threshold. Burst windows closer than 20 ms were considered as the same burst and merged. Isolated bursts shorter than 25 ms were not considered. Trial-by-trial windows of EMG activity with the automatically detected bursts were visually inspected by the experimenter to check for misidentifications by the algorithm. Onsets and offsets of activity were manually corrected when necessary. Furthermore, noisy trials were manually rejected after visual inspection. The peak EMG response of a trial was defined as the peak amplitude within this window. Peak amplitudes lower/greater than 1.5 interquartile ranges over the first/third quartile within individual participant distributions were considered artifacts, and the corresponding trial was rejected. Time series of within-trial EMG activity were time-locked to peak responses and epoched at ± 500 ms around this point. Note that data from nine participants was excluded from the EMG analysis due to poor signal quality (remaining sample: $N = 51$, $n = 27$ females, $n = 24$ males; mean age ± SD = 23.88 ± 4.79).

### Subjective evaluation task.
2D snapshots of the scenes presented in the virtual environment were used for this task. We created images representing each pair of angry-neutral avatars (32 dyads). The task was implemented in MATLAB[66] (R2022b) using Psychtoolbox[73] (v3.0.18). A typical trial (Supplementary Fig. S13) began with a 0.5 s long gray screen followed by a superimposed white fixation cross with a duration randomly sampled from a uniform distribution between 0.8 and 1.2 s. Once

the fixation cross disappeared, the scene was presented for 1 s. The vertical frame containing the arrows in the original task was rotated 90 degrees in a horizontal position, with the arrows pointing to the left or right elevator. After the scene, a screen with a visual analog scale asked participants how much they would like to be in the elevator pointed by the horizontal arrow, ranging from "*Pas du tout*" ("*Not at all*") to "*Vraiment beaucoup*" ("*Very much*"). To respond, participants had to click and drag the cursor on the scale before confirming the response by clicking on the "*Continue*" button, which ended the trial. Based on the position on the scale, responses were attributed a score between 0 and 100. Each possible pair of avatars was presented twice, once with an avoidance situation (i.e., the arrow pointing to the elevator with the neutral face) and once with an approach situation (i.e., the arrow pointing to the elevator with the angry face). This resulted in 64 trials. The order of stimulus presentation and the direction of the arrow (i.e., approaching or avoiding the elevator with the angry avatar) were randomized across trials. The task was preceded by a brief training session of six neutral trials (three pairs of females and three pairs of male avatars) to familiarize participants with the task. Due to a technical issue, data from this task is available for a sub-sample of participants (46 individuals, 28 females; mean age ± SD = 23.24 ± 4.62).

### Procedure

The day before the experiment, participants were asked to complete the LSAS, the STAI-Trait, the BIS-BAS, the PHQ-8 and the PANAS scales via Qualtrics (Qualtrics, Provo, UT). On the day of the experiment, participants were asked to sign an informed consent upon arrival and to complete the STAI-State scale. Next, and only in Experiment 2 and 3, an experimenter attached the ECG electrodes, and a five-minute resting baseline was collected. Note that ECG data were collected in Experiment 2 only to assess the feasibility of such recordings in preparation for Experiment 3. This process helped us identify a technical issue with trigger alignment which we solved for the next experiment. In Experiment 3, in addition to the ECG electrodes, EMG electrodes were attached to participants' hands. Participants were then introduced to the equipment (VR HMD and controllers) and received verbal instructions on the task from the experimenter, who was not blind to the experimental conditions or the study hypothesis. Finally, the experimenter assisted participants in putting on the equipment and allowed them to adjust the headset to maximize comfort. Participants sat on a chair with their arms resting on a table throughout the task. Once they were ready to start, the experimenter left the room. After the experiment, participants completed the Simulator Sickness Questionnaire and answered some debriefing questions. In Experiment 3 only, participants also performed the Subjective Evaluation task (see below). The entire procedure took approximately 75 minutes in Experiment 1, 90 in Experiment 2 and 100 in Experiment 3.

### Statistical analysis (Experiments 1, 2, and 3)

**Model-free behavioral analyses.** Model-free statistical analyses were conducted using R[74] (v 4.1.1) and RStudio[75] (v 2023.6.2.561). Most analyses were performed on Threat trials (i.e. 64 trials, 32 per Condition). We considered trials with RTs below 150 ms as anticipations[76] and excluded them from the analyses. Following this rule, in Experiment 1 we dropped seven trials from the full dataset (0.18% of the total), one trial in Experiment 2 (0.05% of the total), and four trials in Experiment 3 (0.1% of the total).

Dichotomous choices (Avoid the angry avatar = 1, Approach the angry avatar = 0) were modeled trial-by-trial through a Bayesian hierarchical logistic generalized linear model (or Generalized Linear Mixed Effect Model or GLMM) using the R package *brms*[77] (v2.16.1). This method estimates regression parameters through Markov Chain Monte Carlo (MCMC) sampling. Three Markov chains were run in parallel with 10,000 iterations each, 2000 of which were used as a warm-up period. We used as priors for parameter estimation a normal distribution with μ=0 and SD = 2.5 (i.e., skeptical priors) in the case of the intercept and other regression coefficients, an exponential distribution with λ = 2 for the standard deviation parameter,

and a Lewandowski-Kurowicka-Joe (lkj) distribution with η = 2 for the correlation matrix. Similar priors have been previously suggested in the case of GLMMs[78]. We considered the Condition (treatment coded: Unpredictable = 0 vs. Predictable = 1) as predictor. Furthermore, because SR and GD processes are supposed to influence behavior at different response latencies[79,80], we also added RTs (grand average centered) and their interaction with the Condition as predictors. For all models we considered the individual subject as a random effect, and we kept a maximum random slope structure[81].

Stepwise model selection was performed by comparing models with different numbers of predictors through their WAIC values. Only the best-fitting model was considered for further analyses. We based inference on group-level parameters' 95% Credible Intervals (95%CrI) defined as the interval between the 2.5th and 97.5th quantiles of the posterior parameter distributions. We considered an effect to be credible if the relative 95%CrI did not contain zero. In addition, we report for each parameter the proportion of posterior samples lower and greater than zero ($p_{<0}$ or $p_{>0}$) and Bayes Factors for parameters different from zero ($BF_{10}$) for credible effects, or Bayes Factors for parameters not different from zero ($BF_{01}$) for non-credible effects. Detailed information about model fitting and selection, complete regression tables and prior sensitivity analyses are provided in Supplementary Methods 3. For analyses with RTs as dependent variables see Supplementary Note 2.

**Drift-diffusion models.** DDMs computationally characterize decision between two options as an evidence accumulation process, which terminates with the response. A simple version of a DDM fits response rate and RTs distributions to estimate three parameters of interest: (1) the drift-rate ($v$), which represents the rate at which evidence accumulates in favor of a response and is positively affected by the value assigned to the choice[82,83]; (2) the boundary separation ($a$), which represents the amount of evidence that needs to be accumulated before executing a response, and thus reflects response caution[84]; (3) the non-decision time ($t0$), which represents the time needed to encode the stimuli and to execute the motor response. In our task: (*a*) an effect on the drift-rate would suggest that contextual information about the predictability of the action-outcome is sampled in order to guide the decision, consistent with a GD process; (*b*) an effect of boundary separation would suggest that participants choose more carefully in one of the two conditions, e.g., because they assign more salience to an action that is contingent with the desired outcome; (*c*) an effect on non-decision time would suggest that participants in one condition spend more time in encoding the stimuli or in implementing a motor response, when they know that their action consistently achieves the desired outcome. To disentangle among these possible effects (or combinations of them), we fitted several DDMs and then compared them to identify which best described participants' behavior.

We used the HDDM package[85] (v0.9.7) in Python (v3.7.13) to fit Bayesian hierarchical DDMs. This method generates estimates of DDMs individual- and group-level parameters through MCMC sampling. For each model three Markov chains were run in parallel with 20,000 iterations each (5000 of which as warm-up period) and a thinning factor of 3. We used the population-informed priors by Matzke and Wagenmakers[86] to fit the models. In all models the starting point bias was kept fixed and equidistant between boundaries. Consistent with our model-free analyses, we modeled only Threat trials. We set avoidance responses from the angry avatar as the upper boundary and approach responses as the lower boundary. We fitted several models, varying the drift-rate, the boundary separation, the non-decision time, a combination of these, or none of them, by Condition, and then compared their DIC and BPIC values[87]. Further analyses were based on the best fitting model (i.e., lowest DIC and BPIC). Due to the limited number of trials in our task, we only estimated regressors (i.e., the Condition effect) at a group level. We based inference on the group-level parameters' 95% Credible Intervals (95%CrI), defined as the interval between the 2.5th and 97.5th quantiles of the estimates' posterior distributions. We also provide the proportion of posterior samples for each parameter lower and greater than

zero ($p_{<0}$ or $p_{>0}$). Detailed information about DDM model fitting and selection can be found in Supplementary Methods 4.

## Statistical analysis specific to Experiment 3: physiological and subjective measures

**Physiological data.** Physiological time series were analyzed using a one-dimensional statistical parametric mapping approach. Through this method, individual time points of physiological signals (iHR or EMG amplitude) from each participant were fitted separately trial-by-trial using a Generalized Linear Model (GLM) with the following formulas (1–3):

$$iHR_{(t)} \text{ or } EMG_{(t)} \sim \beta_{0(t)} + \beta_{condition(t)} + \beta_{threat(t)} + \beta_{condition*threat(t)} + \beta_{RT} \quad (1)$$

$$iHR_{(t)} \sim \beta_{0(t)} + \beta_{condition(t)} + \beta_{outcome(t)} + \beta_{condition*outcome(t)} + \beta_{RT} \quad (2)$$

$$EMG_{(t)} \sim \beta_{0(t)} + \beta_{condition(t)} + \beta_{response(t)} + \beta_{condition*response(t)} + \beta_{RT} \quad (3)$$

with t being the time-point, representing 100 ms in the case of iHR (interpolated at 10 Hz) and 1 ms in the case of EMG (sampled at 1 kHz). In the models: $\beta_{0(t)}$ represented the intercept, $\beta_{condition(t)}$ represented the effect of the Condition (Unpredictable = −0.5; Predictable=0.5) on the iHR or EMG amplitude time point, $\beta_{outcome(t)}$ the effect of the Outcome (Threat = −0.5; Safe=0.5), $\beta_{response(t)}$ the effect of the Response (Approach the angry avatar = −0.5; Avoid=0.5), $\beta_{threat(t)}$ the effect of Threat (Neutral trial = −0.5; Threat trial=0.5), $\beta_{RT}$ the effect of Response Times (grand average centered), $\beta_{condition*outcome(t)}$ the Condition by Outcome interaction, and $\beta_{condition*response(t)}$ the Condition by Response interaction, $\beta_{condition*threat(t)}$ the Condition by Threat interaction. Note that when testing for the effect of Threat, we could not simultaneously test for the effect of Response, as there were no approach or avoidance responses in Neutral trials. Therefore, for each physiological signal, we fitted different GLMs with different datasets, namely the GLM including the Response effect only on Threat trials, and the GLM including the Threat effect on the full datasets. Before fitting the models, both iHR and EMG data were z-scored within participants using the mean and standard deviation of the concatenated trial data.

We tested the significance of the effects obtained using a cluster-based permutation approach[88]. First, we computed t-values against zero for the beta coefficients for each regressor and each time point. Second, we applied two thresholds to the resulting t-values series at the 2.5th and 97.5th most extreme values of a t-distribution ($p_{thres} = 0.05$). Clusters of effects were defined as contiguous points exceeding these thresholds for more than 100 ms for ECG and 20 ms for EMG. The significance of these clusters was tested by permuting the series of beta coefficients 10,000 times with vectors of zeros (i.e., no effect) for a random number of participants. For each permutation, we extracted the sum of the t-values ($t_{mass}$) within each detected cluster to create a 'null' distribution of t-masses. Finally, the t-masses for each observed cluster were compared to this null distribution to compute non-parametric cluster-corrected p-values ($p_{corr}$) as the proportion of the absolute value of the null distribution that exceeded the absolute value of the observed t-mass. Clusters with a $p_{corr} < 0.05$ were considered significant.

**Data from subjective evaluation task.** First, we conducted a two-sided paired t-test on the mean reported value between the approach and the avoidance situations (i.e., with the arrow pointing toward or away from the angry avatar). The assumption of normality was formally verified through a Shapiro-Wilk test, and the assumption of homogeneity of variance was formally verified through a Levene's test.

Second, since the subjective evaluation task presented the same dyads of stimuli as in the VR task, we were able to compute a per-dyad Subjective Value (SV) score as the difference between the avoidance and the approach value from the angry avatar presented in a given trial of the VR task

(originally from −100 to 100, then scaled between −1 and 1 to facilitate model fitting). Thus, negative values indicated a preference for approaching the angry avatar over avoidance, positive values indicated a preference for avoidance, and zero indicated indifference. These per-dyad values were then fitted on a trial-by-trial basis to a Bayesian hierarchical logistic model predicting avoidance decisions (see below *Effect of Heart Rate and Subjective Value on Avoidance Decisions*).

**Effect of heart rate and subjective value on avoidance decisions.** To test the hypothesis that heart rate deceleration sustains avoidance decision-making by facilitating the integration of subjective values into the decision[60], we computed a trial-by-trial measure of cardiac deceleration by taking the maximum iHR change from baseline value in the cluster of beta parameters significantly lower than zero on the intercept of our GLM (see *Results - Electrocardiography*). Heart rate change values lower/greater than 1.5 interquartile ranges over the first/third quartile within individual participants' distributions were considered artifacts, and the corresponding trial rejected. iHR change values were then z-scored within participants. We then fitted a Bayesian hierarchical logistic model predicting avoidance decisions and including as predictors the Condition, RTs (as in the best fitting model in the previous analyses, see *Results - Avoidance Rate*), iHR change, Subjective Value (SV; see above for a description) and their interactions. Importantly, because the Subjective Evaluation task was only administered to a sub-sample of participants, and since the ECG data for one of these participants was rejected due to low quality of the recording, this model was fitted on 45 individuals ($n = 28$ females, $n = 17$ males; mean age ± SD = 23.31 ± 4.65).

## Finite mixture modeling (FMM)

A limitation of common statistical models is that they assume homogeneity of behavior, with inter-individual differences typically interpreted as variance around a central tendency. Nonetheless, heterogeneity in free choice tasks[34,89], and especially in avoidance behavior, is the norm in both animal and human populations[90–93]. This seems relevant in the current study's case and especially in the interpretation of our behavioral results. Specifically, some participants may avoid above-chance in both conditions and with no difference between them, irrespective of action-outcome predictability, in an SR fashion. Conversely, other participants may learn from outcomes and avoided at chance level in the unpredictable condition and significantly more in the predictable condition, in a GD fashion. To address this possibility and explore possible latent classes in our sample, we used Finite Mixture Modeling (FMM). Mixture models assume that data are sampled from a mixture of generating processes, or different populations. This method allows for data-driven clustering within datasets without any a-priori knowledge.

To improve the power of the finite mixture modeling algorithm, we performed this clustering by combining the samples from Experiments 2 and 3, which employed the same task. The combined sample of Experiments 2 and 3 consisted of 90 participants ($n = 47$ females, $n = 43$ males; mean age ± SD = 23.14 ± 4.2). We implemented FMM using the R package *flexmix*[94] (v.2.3-19). We performed a stepwise mixture model fitting of the previously identified best fitting model in Experiments 2 and 3 (i.e., response predicted by Condition, RTs, and their interaction, see *Results – Model-Free results on avoidance rate*) considering an increasing number of clustering components (i.e., latent classes) from 1 to 5. The best-fitting clustering for each number of components is identified by an Expectation-Maximization (EM) algorithm that optimizes parameters' maximum likelihood over 1000 iterations. We compared models with the best fitting clustering for each number of components using Bayes Information Criterion (BIC), which has been shown to be optimal for comparing mixture models[95]. Finite mixture modeling revealed that two latent classes best explained our data (see Supplementary Fig. S14). Based on the behavioral pattern presented by these classes (see *Results – Comparing GD and SR latent classes – Model-Free results on avoidance*

*rate*), we will refer to them as 'GD Class' ($n = 61$, 17 from Experiment 2 and 44 from Experiment 3) and 'SR Class' ($n = 29$, 13 from Experiment 2 and 16 from Experiment 3).

When considering the effect of latent classes estimated by FMM, Bayesian hierarchical logistic models on approach and avoidance choices were first fit including the Class as a between-group fixed effect (treatment coded: SR Class = 0, GD Class =1). As a robustness check, we also fit models separately for each latent class, obtaining overall the same results (see Supplementary Table S25).

Concerning DDM analyses by class, we allowed the drift-rate (v), boundary separation (a), and non-decision time (t0) to vary by Class. Furthermore, to improve the interpretability of the results, we adopted a cell design matrix, thereby estimating a parameter (v, a, or t0) for each Condition and Class (instead of effects on an intercept as in the previous analyses).

For ECG and EMG data, we rerun the previously described GLM models, separately for each FMM class. Because ECG and EMG data were only available for participants in the third experiment, this analysis included only that sample but retained the classification from the algorithm run on Experiments 2 and 3 combined.

## Reporting summary

Further information on research design is available in the Nature Portfolio Reporting Summary linked to this article.

## Results
### Results (Experiments 1, 2, and 3)

**Model-free results on avoidance rate.** Across experiments, model comparison revealed the best fitting model to be the one including the main effects of Condition and RT. As expected, participants avoided the angry avatar credibly more in the Predictable than in the Unpredictable condition [Exp1: p(Avoid|Pred.)= 0.726, p(Avoid|Unpred.)= 0.647; Exp2: p(Avoid|Pred.)= 0.659, p(Avoid|Unpred.)= 0.599; Exp3: p(Avoid|Pred.)= 0.613; p(Avoid|Unpred.)= 0.574, see Supplementary Table S48 for avoidance rates and median RTs across Conditions and Experiments] as reflected in a credible model Condition effect (Exp1: $\beta = 0.475$, $p_{<0} = 0$, 95%CrI = [0.253, 0.699], $BF_{10} = 128.297$; Exp2: $\beta = 0.303$, $p_{<0} = 0.007$, 95%CrI = [0.065, 0.539], $BF_{10} = 0.927$; Exp3: $\beta = 0.188$; $p_{<0} = 0.015$; 95% CrI = [0.019, 0.360]; $BF_{10} = 0.372$). However, participants still avoided credibly more than chance in the Unpredictable condition as revealed by a credible model Intercept (Exp1: $\beta = 0.703$, $p_{<0} = 0$, 95%CrI = [0.519, 0.896], $BF_{10} = 5.645e + 14$; Exp2: $\beta = 0.416$, $p_{<0} = 0$, $BF_{10} = 29.969$, 95% CrI = [0.206, 629]; Exp3: $\beta = 0.301$; $p_{<0} = 0$; 95%CrI = [0.152, 0.450]; $BF_{10} = 81.860$, Fig. 1B), and confirmed by a two-sided Wilcoxon test against chance (Exp1: $V = 1450.5$, $p < 0.001$, $d = 0.893$, 95%CI$_d$ = [0.640, 1.210], $BF_{10} = 3.1428e + 06$; Exp2: $V = 340.5$, $p = 0.002$, $d = 0.678$, 95% CI$_d$ = [0.330, 1.130], $BF_{10} = 37.446$; Exp3: $V = 1166.5$, $p < 0.001$, $d = 0.502$, 95%CI$_d$ = [0.280, 0.740], $BF_{10} = 90.432$). Of note, although the 95% credible interval for the Condition effect did not include zero in Experiments 2 and 3, the evidence for this effect represented by the Bayes Factor was still low to null. This was likely because the difference between Conditions depended on RTs, with a credible Condition by RTs interaction (Exp2: $\beta = 1.476$, $p_{<0} = 0$, 95%CrI = [0.726, 2.195], $BF_{10} = 182.234$; Exp3: $\beta = 1.115$; $p_{<0} = 0.001$; 95%CrI = [0.458, 1.756]; $BF_{10} = 35.638$, Fig. 1D, E). Interestingly, the main effect of RTs was non-credible in Experiment 2 (95%CrI = [−0.159, 1.019], $BF_{01} = 3.044$), but credible in Experiment 3 ($\beta = 0.714$; $p_{<0} = 0.002$; 95%CrI = [0.229, 1.198]; $BF_{10} = 6.226$). This suggested that slower RTs were associated with a higher probability of avoiding the angry avatar in the Predictable condition, but not (or to a lower extent in Experiment 3) in the Unpredictable one. Finally, in Experiment 1, we found a credible main effect of RTs ($\beta = 1.050$, $p_{<0} = 0$, 95%CrI = [0.571, 1.544], $BF_{10} = 288.812$), but not a credible Condition by RTs interaction (95%CrI = [−0.467, 1.013], $BF_{01} = 4.841$). For a discussion about the differences across experiment on the RTs effects see Supplementary Discussion 1.

**Drift-diffusion model.** Overall, model comparison revealed that the best fitting model included the effect of Condition on drift-rate, boundary separation and non-decision time, with the exception of Experiment 2, where the effect on boundary was not included in the winning model (Fig. 2D). Posterior parameter testing showed that participants exhibited a credibly greater drift-rate toward avoidance decisions in the Predictable compared to Unpredictable condition (Condition effect: Exp1: $\beta = 0.344$, $p_{<0} = 0$, 95%CrI = [0.226, 0.460]; Exp2: $\beta = 0.303$, $p_{<0} = 0$, 95%CrI = [0.134, 0.472]; Exp3: $\beta = 0.185$, $p_{<0} = 0.003$, 95%CrI = [0.056, 0.312]). However, the drift-rate in the Unpredictable condition was still credibly above zero (Intercept: Exp1: $v = 0.575$, $p_{<0} = 0$, 95%CrI = [0.398, 0.753]; Exp2: $v = 0.384$, $p_{<0} = 0$, 95%CrI = [0.161, 0.605]; Exp3: $v = 0.298$, $p_{<0} = 0.001$, 95%CrI = [0.131, 0.468]), suggesting enhanced evidence accumulation for avoidance decisions even when information about a predicted outcome was not available.

Second, the non-decision time was not credibly different between conditions in Experiments 1 and 2 (Condition effect on t0, Exp1: $\beta = 0.005$, $p_{<0} = 0.085$, 95%CrI = [−0.002, 0.012]; Exp2: $\beta = 0.007$, $p_{<0} = 0.033$, 95% CrI = [−0.0004, 0.015]), but it was credibly shorter in the Predictable compared to Unpredictable condition in Experiment 3 ($\beta = −0.011$, $p_{>0} = 0$, 95%CrI = [−0.017, −0.006]), although both effects were small.

Finally, we found that the response boundary was slightly credibly higher in the Predictable condition compared to the Unpredictable condition in Experiments 1 and 3 (Condition effect on a, Exp1: $\beta = 0.048$, $p_{<0} = 0.010$, 95%CrI = [0.009, 0.089]; Exp3: $\beta = 0.035$, $p_{<0} = 0.019$, 95% CrI = [0.002, 0.068]), indicating a greater response caution under predictability (Fig. 2A–C).

### Results specific to Experiment 3

**Subjective Evaluation Task.** A two-sided paired t-test on the mean reported values on the Subjective Evaluation Task revealed that participants significantly preferred to avoid (mean value ± SD = 65.231 ± 20.478) rather than approach angry avatars (27.704 ± 20.490; $t_{45} = −14.105$, $p < 0.001$, $d = −2.08$, 95%CI$_d$ = [−2.71, −1.71], $BF_{10} = 1.357e + 15$; Supplementary Fig. S13).

**Electrocardiography.** Cluster-based analysis of event related cardiac responses revealed two significant clusters of beta coefficients for the stimulus-locked iHR series (Supplementary Fig. S15). The beta coefficients for the intercept (i.e., the grand average HR activity) showed a significant negative cluster ($t_{mass} = −214.635$, $p_{corr} = 0.014$), beginning approximately one second before the doors opened and extending approximately 2.5 s after. These negative coefficients indicated a cardiac deceleration around the time of the response in both Conditions and independent of Outcome and RTs. Furthermore, the coefficients for the Condition effect presented a significant negative cluster ($t_{mass} = −188.635$, $p_{corr} = 0.005$), beginning approximately 0.5 s after the doors opened and ending approximately 3 s after the end of the response time. Hence, participants presented a greater and longer lasting cardiac deceleration around the response in the Predictable than Unpredictable condition (Fig. 3A). Supporting the threat-related nature of this effect, cardiac responses did not significantly differ between Conditions in neutral trials (Supplementary Fig. S16).

Importantly, the fact that the cardiac deceleration began before the stimulus onset (Fig. 3A) may be related to the onset of the jittered time interval in anticipation of the doors opening (as participants may already be beginning to prepare the response). Therefore, as a robustness check, we repeated the same analysis described above on time series centered on the beginning of the jittered interval. These analyses yielded the same results.

**Effect of Heart Rate and Subjective Value on Avoidance Decisions.** To explore the hypothesis that cardiac deceleration enhances subjective value integration into choice probability, we fitted a Bayesian hierarchical logistic model predicting avoidance choices by Condition, RTs, maximum iHR change from baseline value around stimulus onset, and

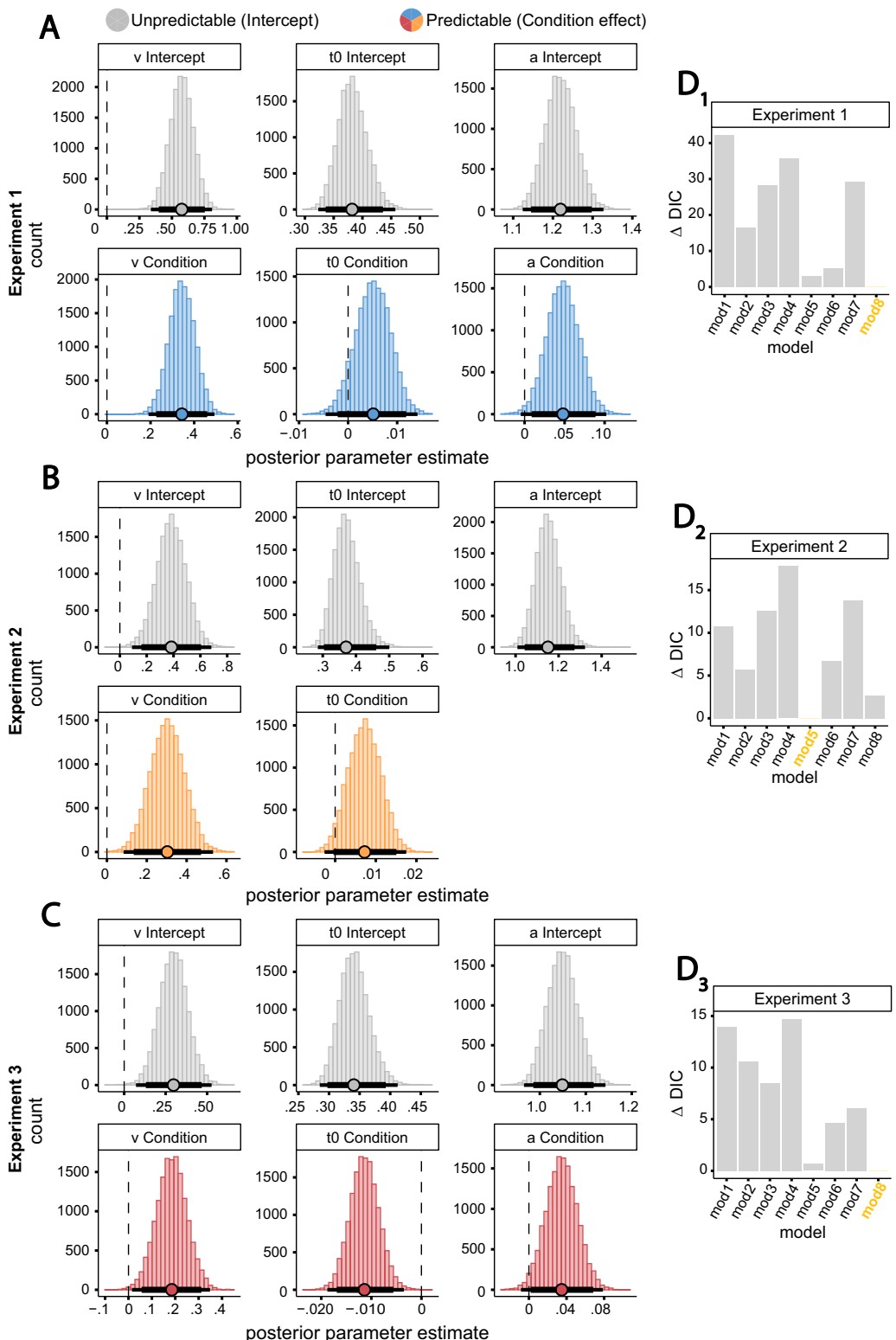

**Fig. 2 | Drift-Diffusion models posterior parameter estimates and model comparison. A** Drift-rate (v), boundary separation (a), and non-decision time (t0) posterior estimate for Experiments 1 (**A**, predictable in blue, n = 60), Experiment 2 (**B**, predictable in orange, n = 30) and Experiment 3 (**C**, predictable in red, n = 60). Intercept: Intercept of the parameter (i.e., the Unpredictable condition). Condition: Predictable condition effect on the parameter. Points: median posterior estimates. Thick and thin black lines: 95% and 99%CrI. **D** DDM model comparison results for each experiment (D1 to 3). ΔDIC (Delta Deviance Information Criterion) represents the DIC difference from the winning model (i.e., lowest DIC value, yellow text on the x axis). mod1: no parameter influenced by Condition. mod2: drift-rate varies by Condition. mod3: non-decision time varies by Condition. mod4: boundary separation varies by Condition. mod5: drift-rate and non-decision time vary by Condition (best fitting model in Experiment 2). mod6: drift-rate and boundary separation vary by Condition. mod7: boundary separation and non-decision time vary by Condition. mod8: all parameters vary by Condition (best fitting model in Experiments 1 and 3).

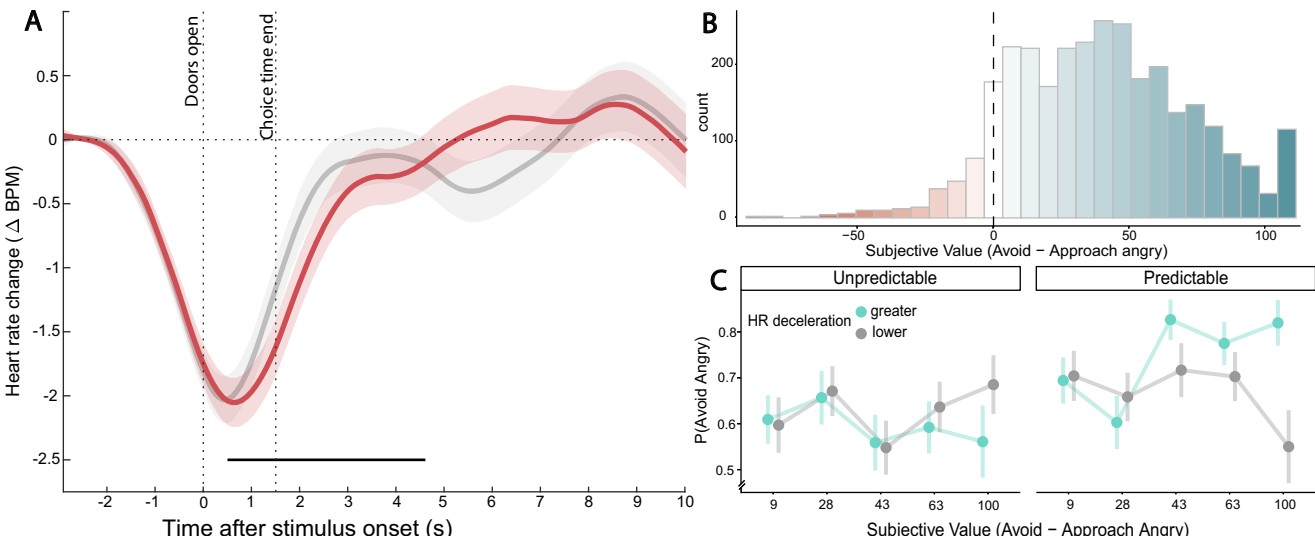

**Fig. 3 | Electrocardiography and subjective evaluation. A** Grand average iHR series (difference from baseline in BPM, Beats Per Minutes) by Condition ($n = 55$). Shaded areas: within subjects 95% confidence intervals. Black thick dotted line: significant cluster of Condition effect in the GLM. Gray: Unpredictable condition. Red: Predictable condition. **B** Histogram of differences between avoid and approach values in the same dyad of avatars, for each stimulus of the subjective evaluation task and each participant ($n = 46$). **C** Proportion of avoidance, P(Avoid Angry), responses by participants' Subjective Value (SV). A credible Condition by HR by SV by RTs four-way interaction emerged ($\beta = -0.931$, $p_{>0} = 0.007$, 95%CrI = subjective value quantile (0.2, 0.4, 0.6, 0.8, 1), by maximum instantaneous heart rate change from baseline (median split; gray: lower change, turquoise: greater change) and Condition ($n = 45$). Note that this plot represents only trials with slower RTs (median split) as the interaction with RT in the regression model showed that this effect is not present with fast RTs. Also note that while we illustrated the effect using median split of iHR change and quantiles of subjective value, the effect in the regression model is on continuous values.

$[-1.685, -0.191]$, $BF_{10} = 3.236$; see Supplementary Table S22), suggesting that greater HR deceleration around stimulus onset was associated with an increased positive effect of SV on the probability of avoiding the angry avatar, with the effect increasing at slower RTs and only when the outcome of the action was predictable (Fig. 3C).

**Electromyography.** Cluster-based permutation analysis did not reveal any clusters of significant effects of our regressors on peak-centered EMG activity (all $p_{corr} > 0.05$), neither when testing the Condition effect nor when testing the Choice effect (i.e., approach vs. avoid angry avatars).

**Comparing GD and SR latent classes**

Finite mixture modeling revealed that two latent classes best explained our data (Supplementary Fig. S14). Considering the behavioral pattern presented by these classes, we will refer to them as 'GD Class' ($n = 61$, 17 from Experiment 2 and 44 from Experiment 3) and 'SR Class' ($n = 29$, 13 from Experiment 2 and 16 from Experiment 3). Although it appeared that the classes were more evenly balanced in Experiment 2 compared to Experiment 3, this difference was not statistically significant ($\chi^2 = 1.838$, $p = 0.175$).

**Model-free analyses of avoidance rates.** A credible three-way Condition by RTs by Class interaction emerged ($\beta = 1.165$; 95%CrI = [0.338, 1.995]; $p < 0 = 0.004$; BF10 = 18.858; see Fig. 4B and Supplementary Table S23). In general, the GD Class, but not the SR one, presented a credibly greater avoidance rate in the Predictable condition compared to the Unpredictable one. Furthermore, the SR Class presented a credibly greater avoidance rate compared to the GD class [SR: P(Avoid|Pred.)= 0.739; P(Avoid|Unpred.)= 0.724; GD: P(Avoid|Pred.)= 0.575; P(Avoid|Unpred.)= 0.515]. This pattern was sustained by the fact that in the GD Class slower RTs were associated with a higher probability of avoidance only in the Predictable condition, whereas this effect was present in both Conditions in the SR Class. Interestingly, in the SR Class, avoidance rate in the Unpredictable condition was credibly greater than chance as reflected by a credible model intercept ($\beta = 0.883$; 95%CrI = [0.718,

1.048]; $p < 0 = 0$; $BF_{10} = 4.696e + 15$), and confirmed by a two-sided Wilcoxon-test against chance ($V = 405$, $p < 0.001$, $d = 1.690$, 95% $CI_d = [1.310, 2.370]$, $BF_{10} = 1.5575e + 07$). Conversely, in the GD class the avoidance rate in the Unpredictable condition was not credibly different than chance level, as reflected by a credible negative Class effect in the model ($\beta = -0.820$; 95%CrI = [$-1.015$, $-0.625$]; $p > 0 = 0$; $BF_{10} = 6.745e + 38$) and confirmed by a two-sided Wilcoxon test against chance ($V = 924$, $p = 0.118$, $BF_{01} = 3.615805$).

**Drift-diffusion model.** When fitting DDMs models considering the latent classes into analyses, we found that the best-fitting model included different drift-rates and non-decision times by Class and Condition, and different boundary separation parameters by Class. Posterior parameter testing on the drift-rate (v) revealed that while participants in the GD Class presented a credibly greater drift-rate in the Predictable condition compared to the Unpredictable condition ($v_{Pred-Unp} = 0.242$; 95%CrI = [0.027, 0.450]; $p_{<0} = 0.014$), such difference was not credible for participants in the SR class ($v_{Pred-Unp} = 0.211$; 95%CrI = [$-0.093$, 0.520]; $p_{<0} = 0.09$). We further tested the Class by Condition interaction by computing the posterior difference of differences [($v_{pred} - v_{unpred}$ in the SR class) $-$ ($v_{pred} - v_{unpred}$ in the GD class)], which was not-credible (95%CrI = [$-0.342$, 0.400]). Finally, participants in the SR Class showed an overall drift-rate credibly higher compared to the GD Class ($v_{SR-GD} = 0.823$; 95%CrI = [0.638, 1.014]; $p_{<0} = 0$; Fig. 4G).

Furthermore, while neither the GD Class nor the SR Class showed a credible difference between Conditions on non-decision time (GD: $t0_{Pred-Unp} = 0.005$, 95%CrI = [$-0.057$, 0.065], $p_{<0} = 0.440$; SR: $t0_{Pred-Unp} = 0.029$, 95%CrI = [$-0.057$, 0.114], $p_{<0} = 0.255$), the GD Class still presented an overall t0 credibly greater than the SR Class ($t0_{GD-SR} = 0.067$; 95%CrI = [0.020, 0.115]; $p_{<0} = 0.003$). However, further testing suggested that while this difference was credible in the Predictable condition ($t0_{SR-GD} = 0.080$, 95%CrI = [0.010, 0.148], $p_{<0} = 0.012$), it was only marginal in the Unpredictable condition ($t0_{SR-GD} = 0.055$, 95%CrI = [$-0.011$, 0.121], $p_{<0} = 0.051$; Fig. 4G).

Finally, we found a not-credible but marginal difference between the classes on the boundary separation, with a slightly greater separation in the SR Class compared to the GD Class ($a_{SR-GD} = 0.110$; 95%CrI = [$-0.003$, 0.222]; $p_{<0} = 0.028$; Fig. 4G).

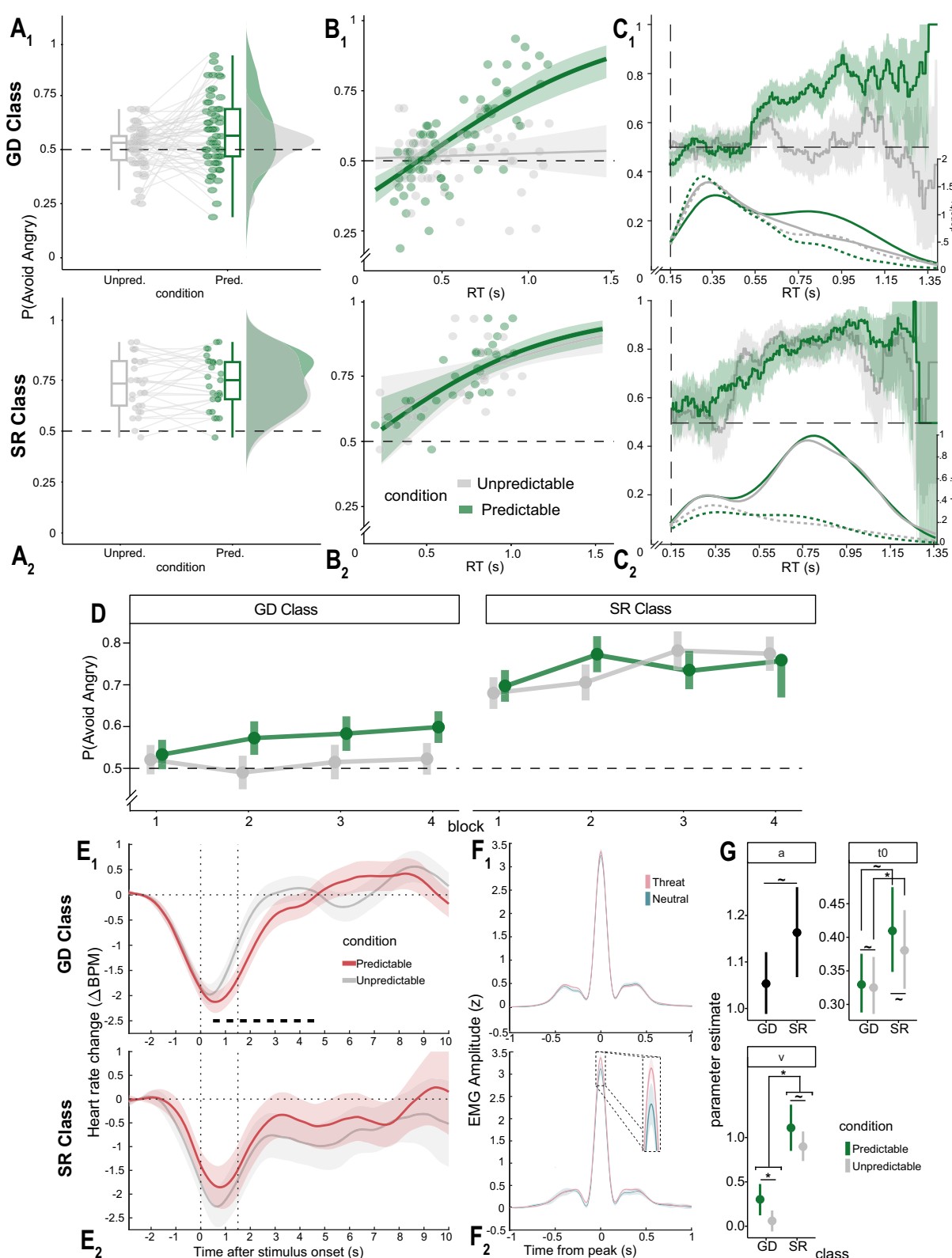

**Electrocardiography**. When considering latent classes into the analyses ($n_{GDclass}=41$; $n_{SRclass}=14$), we found a significant cluster for the estimated intercept around the stimulus onset for both Classes ($t_{massGD} = -177.773$, $p_{corrGD} = 0.008$; $t_{massSR} = -113.153$, $p_{corrSR} = 0.024$), consistent with the full sample. However, while we still found a significant cluster for the Condition effect around the end of the response time for the GD Class ($t_{massGD} = -169.467$, $p_{corrGD} = 0.006$), this

effect was not significant for the SR Class ($t_{massSR} = -58.570$, $p_{corrSR} = 0.053$; Fig. 4E).

**Electromyography**. When considering latent classes into the analyses ($n_{GDclass}=36$; $n_{SRclass} = 15$), we found a significant cluster for the effect of Threat (Threat vs. Neutral trials) at the peak of the EMG activity for the SR Class ($t_{massSR}=477.233$, $p_{corrGD} = 0.032$, length=102 ms), but not for

**Fig. 4 | Main results from classes analysis.** GD Class: Goal Directed Class. SR Class: Stimulus-Response Class. **A** Proportion of avoidance choices across conditions. Box-plots represent first, second (median) and third quartiles. Whiskers are drawn within the 1.5 interquartile range. **B** Predicted proportion of avoidance choices as a function of RTs (solid lines). Shaded areas: 95% credible intervals. Single points: observed avoidance rates over median RTs for each participant. **C** Observed proportion of avoidance responses, P(Avoid Angry), as a function of RTs within a 100 ms sliding window. Shaded areas: 95% bootstrapped Confidence Intervals. On the bottom: defective density distribution (i.e., weighted by the relative frequency of responses) of RTs for avoidance (solid) and approach (dotted) responses. Black dashed lines: chance level. **D** Proportion of avoidance responses by Condition (Unpredictable vs. Predictable) by Class and Experimental block. Vertical lines: within subjects 95% confidence interval. **E** Grand average iHR series (difference from baseline in BPM, Beats Per Minute) across conditions for the GD (n = 41) and SR (n = 14) classes. Thick dotted line: significant clusters for the Condition effect in the GD class. Vertical dashed lines: stimulus onset (doors' opening) and end of response times. Thick lines: grand averages. Shaded areas: within subject 95% confidence intervals. **F** Grand average EMG peak-centered amplitude for Neutral (teal) and Threat (pink) trials for the GD (n = 36) and SR (n = 15) classes. Thick dotted line: significant cluster of Threat effect in the SR class. Shaded areas: within subject 95% confidence intervals. A–C and E–F₁: Results for the GD class (n = 61). A-C and E-F₂: Results for the SR class (n = 29). **G** Posterior parameter estimates for DDM drift-rate (v), boundary separation (a) and non-decision time (t0) by Class and Condition. Points: median estimate. Lines: 95%CrI. Note that green represents the Predictable Condition where the sample is combined between Experiments 2 and 3 (**A–D, G**), while red represents the Predictable Condition where the sample is only from Experiment 3 (**E**). Unpred Unpredictable condition, Pred Predictable condition, RT Response Time.

the GD Class. As illustrated in Fig. 4F, unlike the GD Class, the SR Class participants presented a significantly stronger EMG amplitude when producing a response in trials in which they were presented with angry avatars compared to trials in which only neutral trials were presented.

## Discussion

Recent evidence suggests that not only stimulus-response (SR) associations, but also rapid and implicit goal-directed (GD) processes play a key role in threat avoidance decisions[7,9,30]. To assess the contribution of SR and GD processes to social threat avoidance, we manipulated outcome predictability of approach-avoidance decisions when facing social threat, by contrasting an experimental condition with predictable action-outcome contingencies with one featuring unpredictable contingencies. Our findings support the central role of GD processes in social threat avoidance, while shedding light on inter-individual variability in avoidance and on the computational and physiological correlates of the decision.

Across three experiments, we found that participants more frequently avoided angry avatars when they could reliably anticipate the outcomes of their actions (predictable condition), compared to when outcomes were stochastic (unpredictable condition), and independently of the specific dyad of avatars presented on a trial (see Supplementary Note 3, but also Supplementary Note 4 for a test on the effect of avatars' sex). This suggests that social threat avoidance is influenced by GD processes, which are sensitive to action-outcome predictability. Intriguingly, the above-chance avoidance rate in the unpredictable condition indicated that SR associations also influence behavior. This combination of GD and SR processes may reflect intra-individual variability, with both processes running in parallel during predictable scenarios[9], promoting threat avoidance. Alternatively, this behavioral pattern may be due to inter-individual variability[96,97], with some individuals relying more on GD and others on SR processes.

Consistently with the latter, we identified two latent classes of participants, which we named GD and SR, respectively. Most participants exhibited a clear goal-directed behavioral pattern (GD Class), with chance-level avoidance in the unpredictable condition and above-chance in the predictable condition. Their avoidance rate increased over time only in the predictable condition (see Fig. 4D and Supplementary Note 5), suggesting that they learned in which condition the action-outcome contingency was predictable and adapted their behavior accordingly. A minority of participants exhibited a stimulus-driven behavioral pattern (SR Class), with no difference in avoidance rate between conditions, and a higher overall avoidance rate compared the GD Class (see also Supplementary Note 6). Consistently, previous studies demonstrated heterogeneous latent patterns in aversive learning in rodents[91,93], as well as in healthy[92] and clinical[90,93] human populations. These studies revealed that in fear conditioning paradigms, while some individuals learn that a stimulus is no longer threatening, consistent with individuals in the GD Class, others exhibit diminished safety learning and maintain generalization of fear responses, consistent with individuals in the SR Class.

Computationally, the difference in avoidance rates between conditions was subtended by an increased drift-rate to avoidance, i.e., enhanced evidence accumulation process, in the predictable compared to the unpredictable condition, specifically in the GD Class (for a meta-analysis of the reliability of DDM parameters, see Supplementary Note 7). In value-based decision-making, the drift-rate reflects noisy value estimates for the alternative actions[98] (here avoidance vs. approach), so it is influenced by the value assigned to each option[21,49,82,83]. This result reinforces the idea that threatening expressions influence action selection by increasing the predicted value of avoidance[21,44,49]. Furthermore, it suggests that in the GD Class, value estimates are influenced by action-outcome predictability, consistent with a GD process.

In contrast, the SR Class showed a higher drift-rate compared to the GD Class, with no difference between predictable and unpredictable conditions. This signaled that value attribution was unaffected by action-outcome predictability, consistent with a SR process. Moreover, participants in the SR Class exhibited slightly higher boundary separation and greater drift-rate across conditions, and longer non-decision time in the predictable than the unpredictable condition, compared to the GD Class. Previous studies found that drift-rate, boundary separation and non-decision time increased in salient emotional contexts[99–101]. Also, in non-socioemotional perceptual decision-making, participants exhibited higher boundary separation and longer non-decision time[102] to improve accuracy by increasing response caution and/or adopting strategic motor slowing[103]. These differences subtend the main effect of class in our model-free analyses, i.e., greater avoidance rate in the SR compared to GD class, irrespective of condition. Participants in the SR class may have been more sensitive to social threat, increasing their avoidance rates and reducing their ability to adapt to environmental contingencies in a goal-directed manner. In line with the predatory-imminence model[17,18], this suggests that the increased reliance on a reactive behavioral policy occurs with increased perceived threat salience and/or decreased perceived distance from the threat. Therefore, inter-individual differences in threat sensitivity determine a shift between the GD and SR controllers when lying in the transition zone that characterizes the post-encounter of an angry or aggressive individual.

Finally, physiological results further clarified the decision processes. Across conditions and groups, we observed a cardiac deceleration around stimulus onset, consistent with previous studies on non-emotional[104–106] and emotional[59,62,107] decision-making. In threatening contexts, a similar cardiac deceleration, observed when facing a well-identified but not imminent threat[107,108], has been interpreted as an autonomic activity facilitating threat responses[109,110]. Here, cardiac deceleration was greater and longer lasting in the predictable vs. unpredictable condition, only in threat trials (see also[111]), and only in the GD Class. This result adds to the literature, by providing evidence that cardiac deceleration is not a mere physiological reaction to threat but is associated with the organization of goal-directed defensive behavior.

An interesting hypothesis is that cardiac deceleration influences approach-avoidance decisions, by favoring the integration of subjective

value into action selection, rather than simply facilitating information gathering and action execution[60,62,112]. Our findings support this hypothesis, as increased cardiac deceleration was linked to better integration of subjective values in avoidance decisions, specifically in the predictable condition and with slower response times. Therefore, we showed that cardiac activity relates to goal-directed decision-making under threat. This claim is further supported by the fact that only the GD Class presented a higher avoidance rate, a higher drift-rate toward avoidance, and a greater and longer lasting cardiac deceleration in the predictable vs. unpredictable condition. In contrast, the SR Class did not show differences in cardiac deceleration between conditions but showed higher EMG amplitude for threat than neutral trials. Such increased muscular response, a sign of action readiness[63–65], suggests a stronger aversion for the scenes with angry avatars, in line with behavioral and DDM findings. However, we found no difference in the subjective evaluation of approach-avoidance outcomes between classes, which suggests cautiousness in this interpretation.

Overall, our results have shown that social threat avoidance is goal-directed for most individuals, though a minority of participants exhibited stimulus-driven responses, possibly due to heightened threat sensitivity. This might seem counterintuitive given previous work suggesting that habitual SR processes and GD processes run in parallel competing for action selection, and that SR may prevail only under strict time constraints[80]. Here, time constraints were not so strict, as participants had 1.5 s to respond. However, no efficient goal-directed process relative to the location of the emotional expression of the avatar could take place in our unpredictable condition, precisely because the action-outcome contingency was stochastic. Therefore, differently from previous tasks, the two processes could not compete and avoidance, if present, could only be SR irrespective of time constraint. On this line, avoidance responses in the SR Class correlated positively with RTs in both conditions, i.e., the longer participants took to respond, the more SR processes influenced behavior. The fact that, in tasks where SR and GD processes compete, SR effects typically appear in early time windows[80] does not mean that SR are short-lived, but rather reflects the time needed for GD processes to win such competition and overcome the SR associations.

Regarding the chance-level avoidance rate in the GD class under the unpredictable condition, an alternative interpretation is that it reflects participants exploring, through goal-directed mechanisms, the impact of their choices on the avatar changes (a stochastic event absent in the predictable condition). In partial contradiction to this interpretation, we found no evidence of learning effects in the unpredictable condition (see Supplementary Note 5). Nonetheless, since nothing could be concretely learned in the unpredictable condition, it is possible that action-outcome unpredictability prompted a shift from an exploitative to an exploratory strategy. This is intriguing, as it suggests that social threat under uncertainty might promote exploratory GD processes, warranting cautious interpretation of our results. Future studies ought to dissociate between the two possible interpretations, i.e., either the absence of GD exploitative or the presence of explorative strategies, for instance by manipulating the response preparation time. Imposing time pressure in an unpredictable context may impede exploratory GD strategies, clarifying whether individuals rely on SR associations. This manipulation would also be pertinent for classical forced-choice approach-avoidance tasks, which are typically used to test SR tendencies and give participants up to two seconds to respond[2,3,13].

Following on from the previous point, while our VR task allowed us to selectively affect GD processes by manipulating action-outcome predictability, it did not allow us to independently manipulate SR processes. To achieve a more fine-grained computational characterization of social threat avoidance, future studies ought to manipulate GD and SR processes orthogonally. One idea, beyond the above-mentioned manipulation of the response time window, is to act on the perceived distance to threat. Following the predatory-imminence model, which suggests that imminent threats are more likely to elicit SR-driven reactions[17,18], a condition could be added where participants are attacked as they enter the elevator with the angry avatar. Recent VR studies employed similar manipulations[113,114],

which could be orthogonalized with outcome predictability and provide a more exhaustive picture of mechanisms involved in social threat avoidance.

Finally, it is worth noting that behavior in the current task can be described in terms of either avoiding the angry avatar or approaching the neutral one. It is debated whether avoidance is driven by appetitive (i.e., reaching safety), aversive (i.e., avoiding threat) motivation, or both[115]. Indeed, approaching safety and avoiding threat are necessarily intertwined, but at the same time appetitive and aversive motivation might be partially dissociable in terms of neural correlates. Indeed, behavior driven by positive vs. negative reinforcers is in part sustained by common brain networks[116,117], but the overlap is not complete, as some brain regions seem to respond selectively to each kind of reinforcer[52,116] and different neurotransmitters may be implicated[118]. Here we do not have further arguments to dissociate between the two possibilities. Nonetheless, the effect of outcome predictability on cardiac responses is conditioned to the presence of angry avatars (i.e., no difference for neutral trials). This suggests that, at least physiologically, the decision process when approaching neutral is not the same when a threat is present or absent in the scene. Future studies should explore threat avoidance without the confound of safety by including trials where participants can refrain from entering any elevator or enter an empty one. In such scenarios, individual social motivations may lead participants to prefer avoiding or approaching the elevator with the avatar, regardless of its emotional expression. This would move the research focus from social threat avoidance to general social avoidance. Nonetheless, due to the relevance of general social avoidance for several psychopathological conditions, from social anxiety to depression, the topic merits further investigation.

## Limitations

The current study is not without limitations. First, although we selected the sample size for each experiment using appropriate power simulations for the condition effect, these experiments may be underpowered with respect to other effects. A larger scale replication would ensure greater power, for example, to identify more latent classes of behavior, providing a finer-grained picture of interindividual differences in social avoidance. Furthermore, although our sample is perfectly matched in terms of sex, we sampled participants by convenience. Future studies should employ random sampling to ensure the generalizability of the present results.

Second, the task involved a limited number of trials (64 Threat trials, 32 per Condition). This is an inherent limitation of VR, as too many trials can cause cybersickness. We addressed this limitation by using hierarchical modeling in our analyses and basing inference on group-level parameters. Nevertheless, we could not obtain individual estimates of the effect of condition on DDM parameters, constraining the exploration of inter-individual differences.

Third, while VR offers a more ecological and realistic experience, it also presents challenges in controlling participants' spatial attention. Here, participants were free to explore the visual scene, potentially inducing variability in the way participants attended to avatars' expressions during the outcome phase. Yet, our analysis of head-tracking measures (Supplementary Note 8) indicated that participants consistently oriented their head towards the avatars during this phase, irrespective of the condition.

Fourth, participants performed the entire experiment seated, to align with previous literature on computerized tasks. While we showed that avoidance decisions can be motivated by GD processes, reactive SR processes may affect covert measures of behavior, such as posture or gait[119,120], during action selection or execution. Future studies should implement kinematics measures to investigate this possibility.

## Conclusions

Using a VR task that manipulates action-outcome predictability of social threat avoidance, we provided behavioral, computational and physiological evidence in three experiments supporting the hypothesis that GD processes are predominant in driving threat avoidance in socio-emotional contexts. Furthermore, the present results shed light on inter-individual variability in

avoidance, identifying two classes of behavior when facing social threat, the most prevalent being GD and the other one being SR.

Our findings offer interesting perspectives for understanding and treating clinical disorders. A widely accepted interpretation of maladaptive approach-avoidance behavior in psychopathology (e.g., social anxiety) relies on the presence of biased and prepotent action tendencies and/or weakened goal-directed processes, unable to override such tendencies[8,121,122]. Therapeutic approaches have aimed to strengthen GD inhibitory processes or instill more adaptive tendencies[123–125]. Nonetheless, although such training was effective in increasing approach tendencies to social stimuli, the effect was short-lived[126]. Our findings suggest that maladaptive social avoidance possibly stems from aberrant value assignment to approach-avoidance actions, so that social contexts that do not systematically lead to threatening outcomes are avoided. Therefore, an alternative therapeutic approach may be to prioritize the reassessment of the values that individuals assign to potential actions, considering the environment's probabilistic structure. This entails enhancing their ability to integrate contextual information and predicted outcomes into action choices, promoting flexible and optimal behavior.

## Data availability
All data are available at the following OSF repository: https://osf.io/rucz9/[45]

## Code availability
Data preprocessing and analysis scripts are available at the following OSF repository: https://osf.io/rucz9/[45]

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

## Acknowledgements

We thank Abhay Koushik and Constance Destais for their help with the creation of 3D virtual stimuli; Constance Destais, Junlian Luo and Lucile Bottein for their help with data collection. We further wish to thank the editor and the anonymous reviewers for their precious help in shaping the final version of the manuscript. This work was supported by INSERM; ENS and the French National Research Agency under Grants ANR-20-CE28-0003; ANR-10-IDEX-0001-02 and ANR-17-EURE-0017 FrontCog. The funders had no role in study design, data collection and analysis, decision to publish or preparation of the manuscript.

## Author contributions

Matteo Sequestro: Conceptualization, Methodology, Software, Formal Analysis, Investigation, Writing—Original Draft. Jade Serfaty. Software, Investigation, Writing—Review and Editing. Julie Grèzes: Conceptualization, Supervision, Project administration, Resources, Funding acquisition, Writing—Review and Editing. Rocco Mennella: Conceptualization, Supervision, Funding acquisition, Writing—Review and Editing.

## Competing interests

The authors declare no competing interests.
