## [Transparent Peer Review file · Communications Psychology]

Social threat avoidance depends on action-outcome predictability

Corresponding Author: Mr Matteo Sequestro

Version 0:

Decision Letter:

Dear Mr Sequestro,

Thank you for your patience during the peer-review process. Your manuscript titled "Social threat avoidance depends on action-outcome predictability" has now been seen by 3 reviewers, whose comments are appended below. You will see that they find your work of some potential interest. However, they have raised quite substantial concerns that must be addressed. In light of these comments, we cannot accept the manuscript for publication, but would be interested in considering a revised version that fully addresses these serious concerns.

We hope you will find the Reviewers' comments useful as you decide how to proceed. Should additional work allow you to address these criticisms, we would be happy to look at a substantially revised manuscript. If you choose to take up this option, please highlight all changes in the manuscript text file, and provide a detailed point-by-point reply to the reviewers.

Editorially, we want to see a clarification of the contribution of this manuscript and the conclusions that can be driven from the current set of experiments. From the reviewers' comments and our own reading, it is evident that the methods do not allow to directly measure and test S-R processes nor to tackle the question of the arbitration between S-R and goal-directed mechanisms in social decision making. We are still convinced that the current data bring an interesting contribution on how the goal-directed processes involved in social decision making, but the manuscript will need to be clear about this aspect. Moreover, we consider that the following aspects will need to be carefully addressed. (1) As highlighted by reviewer#3 and reviewer#2 we want to see a better rationale for the methodological choices (e.g., the choice and implementation of the different measures, the choice of having neutral avatars trials). (2) Reviewer#1 and reviewer#2 also raise the possibility that participants could develop learning strategies during the experiments. We think this point is particularly important, because it is possible that the reduction of social threat avoidance in the uncertain condition could derive from goal-directed strategies aiming at avoiding the unpredictable threat (e.g., entering in the elevator with the angry avatar, hoping that that it will switch). (3) Reviewer#2 raises several very important concerns the analysis (especially the exploratory ones) and also has several suggestions on how to address them, we want to see them implemented.

I am attaching a checklist that details critical reporting requirements for the revised manuscript. Please attend to each item and ensure your manuscript is fully compliant. We are requesting that your manuscript aligns with these requirements as this facilitates the evaluation of your manuscript, reducing delays in re-review and potential future acceptance. If your revised manuscript is not aligned with these requests on major issues, such as those concerning statistics, it may be returned to you for further revisions without re-review. Additional information can be found in our style and formatting guide Communications Psychology formatting guide.

If the revision process takes significantly longer than five months, we will be happy to reconsider your paper at a later date, provided it still presents a significant contribution to the literature at that stage.

Please use the following link to submit your

- revised manuscript,
- point-by-point response to the referees' comments,
- cover letter (as a separate document),
- the Editorial Policy Checklist (see below),
- the Reporting Summary (see below), and
- the completed Editorial Request Table (attached):

Link Redacted

Thank you for the opportunity to review your work.

Best regards,

Eva R. Pool

Eva R. Pool, PhD
Editorial Board Member
Communications Psychology
orcid.org/0000-0001-5929-1007

REVIEWER EXPERTISE:

Reviewer #1: Dual learning models
Reviewer #2: Computational models of social mechanisms
Reviewer #3: Social and affective psychology

REVIEWER REPORTS:

Reviewer #1 (Remarks to the Author):

The authors conducted three studies to investigate the influence of stimulus-response (SR) and goal-directed processes (GD) in social threat avoidance. In an elegant paradigm, participants are faced with a binary choice between two elevators that each have an avatar inside. On threat trials, one of the avatars has an angry face and the other a neutral face. Importantly, a cue indicates to participants if their choice leads to a predictable or unpredictable outcome. In the predictable condition, avoiding or approaching the angry avatar leads to this outcome, whereas this is stochastically determined in the unpredictable condition. Results indicate that participants avoided angry avatars more when the outcome of their choice was predictable, which could be explained by more efficient evidence accumulation in that condition. In addition, physiological data showed that value integration was facilitated by greater cardiac deceleration at the time of choice in the predictable context. Another interesting finding is that participants could be clustered into two groups, one which avoided angry avatars only in the predictable condition, and another which avoided angry avatars in both conditions.

This is a nice series of studies with interesting results for various areas and a well-written manuscript. The analysis was not preregistered, but the predictions follow from the theory. Data and code are available. Below are some of my comments:

1. The authors hypothesized "if social threat avoidance is driven by SR processes, then there should be no difference in avoidance rates between conditions (as SR processes are insensitive to the predicted outcome)." However, recent work shows that SR processes may only determine behavior under strict time constraints and are quickly controlled by GD processes (Hardwick et al., 2019). While the authors took into account RTs, observed RTs may be different from response preparation time manipulation. I'm not sure if the hypothesis is applicable in the current study design as participants have up to 1.5 seconds to respond. Thus, while the authors have convincingly shown that social threat avoidance is strongly influenced by GD processes, I wonder to what extent the current experiments can speak to initial SR tendencies. In support of this, GD avoidance in the predictable increased with more RTs.
2. Related to the first comment, participants may develop goal-directed strategies in this experiment. To this end, it could be interesting to compare the first block to the last block, to see if avoidance is more common in the first block when goal-directed strategies to selectively avoid the angry avatars in the predictable condition may not be as strong.
3. To determine if avoidance is selected above chance level, the authors report the intercept. However, in particular in experiments 2 and 3, the figures (2B and 3B), it looks like the proportion of choosing avoidance in the unpredictable condition may not be different from chance (.5). I think one sample t-tests comparing the proportion to avoid against chance would perhaps be meaningful. On that note, it would be helpful if the authors reported means for the results.

4. The authors state “To test whether a behavior is GD, a classical approach consists in manipulating the outcome value of a response (e.g., by outcome devaluation or contingency degradation; Balleine & Dickinson, 1998).” Strictly speaking, the contingency degradation does not manipulate the outcome value but instead response-outcome contingency.
5. Moreover, they state “Indeed, GD behavior, but not SR, is sensitive to the prediction of the value of the outcome of the chosen action, as well as to the predictability of the association between actions and outcomes (Balleine & Dickinson, 1998; De Houwer et al., 2023).” I believe that the term “contingency” would be more precise than “predictable”. In a contingency degradation paradigm, the reinforcer may be predictable but responding is not instrumental. I think the point is that in the “unpredictable” condition responses are “not instrumental”.
6. Finally, they state “Specifically, when an action reliably leads to the same consequence (controllable action-outcome context), GD processes should be privileged as they are more optimal.” However, some work indicates the opposite, where SR are stronger when reinforcers are predictable because less attention needs to be paid in a stable context (Bouton, 2021). Here again, this may be due to the word predictable rather than instrumental.

Reviewer #2 (Remarks to the Author):

In this paper, the authors investigate the balance between stimulus-response associations (SR) and goal-directed processes (GD) during an approach-avoidance task in order to shed light on the computational mechanisms by which people learn and respond to social threat. Approach-avoidance was tested in a virtual reality social context where participants had to decide between entering an elevator with a neutral-face avatar or an elevator with an angry-face avatar. In one condition, the avatars initially present in the elevator were independent from the avatars that would ultimately walk in at the end of the trial (unpredictable) while in another condition the same avatars would stay on from the beginning (predictable). The authors found that participants were generally likely to avoid the elevator with the angry avatar, and that this tendency increased in the predictable condition, consistent with an increased reliance on GD processes when the environment is predictable. This finding was also accompanied by changes in drift-diffusion model (DDM) parameters, particularly the drift rate, as well as changes in heart rate deceleration, and was found to vary across individuals.

I generally really enjoyed reading this paper and the progression between the three experiments. I do believe that it makes a novel and substantial contribution to the topic of social avoidance behavior. However, I do have quite a lot of comments, including a few major concerns, that I'd like to see addressed before recommending the manuscript for publication.

Major comments:

1) The study rationale is mainly grounded in the SR vs GD dichotomy, but the specific analysis methods and interpretation of the findings often fails to make the link with these processes. A few examples:

1a) I understand that SR is assumed to be an automatic response and therefore should not be influenced by the predictability of outcomes – however, seeing a change in behavior when increasing predictability does not mean that SR is not driving behavior. SR could be driving behavior together with GD, and the ‘level’ of SR remains the same when predictability increases. It seems key to obtain an independent behavioral measure of SR and/or manipulate SR in an independent way from GD.

1b) Related to the above, the DDM makes a clear prediction that changes in drift rates are reflective of changes in GD processes, but nothing in the model seems to be able to independently quantify or characterize SR processes. My understanding of DDMs is that they also usually include a starting point parameter – which in this case seems like it would best capture the biased preference to avoid the angry avatar (and could also constitute the signature of SR processes?)

1c) In the GLM on avoidance decisions – could the intercept be interpreted as a signature of SR processes? If so, could the authors find a way to actually assess the relative contribution of SR vs GD processes in the two conditions (maybe by comparing the standardized intercept with the standardized effect of condition) so their hypothesis can be more directly tested?

2) I have a bit of a general concern with the way avoidance is implemented in the main task and the subjective evaluation task. I understand that participants are either choosing the neutral over the angry avatar, or that they are rating how much they'd like to enter the elevator with the neutral avatar, but both of these behavioral metrics would be more directly described as approaching the neutral avatar (rather than avoiding the angry avatar). Could the findings be different in any way if “true” avoidance had been measured (i.e. the tendency to not approach at all)? This would be worth discussing.

3) I understand that most of the model-free and modelling analyses focus on Threat trials as they aim to analyze the tendency to approach vs avoid the angry avatar. That said, what was the rationale for having neutral trials in the task if those were not considered in any of the analyses? I assumed the goal of the neutral trials may have been to control for any effect of the identity of the avatar beyond the facial expression, which would be worth determining. Given that EMG amplitude was found to differ between threat and neutral trials, it would be great if there was a way to assess any differences in behavior as well (e.g. with regards to RTs, subjective valuation, and/or influence of avatar identity).

4) The motivation for the specific analyses run with the self-report questionnaires is unclear – it would be great if the authors could better motivate the rationale for these analyses in the introduction with appropriate literature reviews on what symptom dimensions is expected to be associated with signatures of GD versus SR processes, thus justifying the subsequent choices of associations being tested. More specifically, what is the rationale for only examining correlations between self-report measures and the condition*RT interaction? Why were pairwise correlations run for this latter analysis but a repeated measures ANOVA for the data from the Subjective Evaluation task? Was correction for multiple comparisons performed (in the case of running the same analysis multiple times with different questionnaires)? And finally, was the multi-collinearity between questionnaires established as well as specificity of the findings? For example, for the findings about the effect of state and trait anxiety on subjective evaluation of approach vs avoidance, was a single ANOVA run with both state and trait

anxiety as predictors (as well as the other scales), or were two separate ANOVAs run? In the first case, this may be problematic given that state and trait anxiety are generally too highly correlated in the population, which would lead to effects in opposite directions (as reported) given that the amount of unique variance is too small to be properly assigned to each measure. In the second case, correction for multiple comparisons should be run as well as ensuring that the effects are robust to controlling for other scales. My recommendations for these analyses would be to first test the association with each self-report with an a priori hypothesized association, correcting for multiple comparisons, then ensure that any significant association is robust to controlling for other self-report measures, other behavioral metrics from the task, as well as demographic information (e.g. age and gender, if available). Alternatively, an option could be to perform some dimensionality reduction on the questionnaire scores (or subscales or items, though for the latter the sample size may be too low) in order to extract symptom dimensions that are less correlated with each other.

5) I find it worrying that there is no main effect of SV on p(avoid angry) in Table D10 or from visually examining Figure 3I. This suggests that people's subjective valuation of the angry vs neutral avatar did not overall influence their choices. Maybe this main effect does not come out in the model in Table D10 because of the model complexity? It would be interesting to test if this main effect is there in a simple model predicting choice from SV only.

6) It's a bit hard to understand the rationale for why heart rate change was used in some analyses (e.g. Figure 3F), but then heart rate seems to be the variable used in predicting avoidance rates (Figure 3I). Is it a faster deceleration or a slower heart rate that's associated with more avoidance in the predictable condition? If deceleration is the key metric, shouldn't a trial-by-trial index of deceleration (rather than minimum heart rate) be used in the GLM from Figure 3I?

7) In the partial discussion sentence on p.32-33 ("Cardiac deceleration was also found to facilitate the integration of subjective outcome values into decisions only in predictable trial"), please refrain to using words like "facilitate", since the causal association between cardiac deceleration and choice was not directly tested. Related to my point above about the distinction between heart rate and deceleration, it would be key to establish that the findings are specific to cardiac deceleration, and not just slow heart rate, in order to make such a claim.

8) The exploratory analyses of heterogeneity are a great addition to the paper, especially to give more insights into individual differences in DDM parameters (which couldn't be reliably estimated) and potentially in self-report symptom measures (although there were no association, but maybe this would change depending on how my point #4 above is addressed). My main concern is that by definition, the plots shown in Figure 4C-D, and corresponding stats on p.35 (avoidance rate paragraph) are somewhat circular. While it's okay to show those illustrations as a sanity check to characterize the behavioral differences between the two clusters and make sure those match the differences from Figure 4B, it's obvious that a 3-way condition*RT*class interaction will emerge given that the clusters were defined based on the condition*RT model, so I would refrain from any statistical report/inference on these findings.

9) The low sample size of the experiments should be noted as a limitation, especially with regards to the heterogeneity and individual difference analyses. Effect sizes of associations between cognitive processes and self-report measures are known to be low, however, the studies were not powered with these low effect sizes in mind.

Minor comments (in rough order of text):

10) From the abstract, it is unclear how the improved performance, increase drift rate and greater cardiac deceleration in the predictable condition helps separate SR versus GD tendencies.

11) What were the participants instructed to in terms of what their goal is in this task?

12) Was there an effect of the gender of the avatars?

13) Thank you for sharing videos of demo trials, this is very helpful in understanding the task – however, I am confused as to why the avatars' faces are blurred in this video? Also, it seems that in the unpredictable trial, when the avatars walk out of the elevator, it is the same avatar that walks back in rather than a new avatar. But maybe it's the blurred face that gives this impression.

14) Is it possible that participants are trying to learn something in this task, especially in the unpredictable condition, whereby even though they are told the new avatars could display an angry vs neutral face with a 50% probability, they may still try to learn whether choosing the elevator with the angry or neutral avatar first has more chance of "attracting" the new angry versus neutral avatar. It may be worth running a quick behavioral check, such that looking at the effect of past trial outcome on subsequent trial choice, or plotting behavior across trials, to ensure that there is no learning effects.

15) p.10: "we also fitted RTs (grand average centered) and their interaction with the Condition as predictors" – this sentence is unclear – saying "we fitted RTs" suggests that you ran a separate GLM to predict RTs from other variables like Condition, but the rest of the sentence suggests that RTs were included as a predictor of choice? I understand from the results section that the latter seems to be what was done, in which case, the sentence in the methods should read "we also added RTs" (rather than fitted).

16) A GLM actually fitting RTs may be interesting (e.g. something along the lines of $RT \sim \text{avoid_choice} * \text{condition}$) – this would be the correct model to test whether RTs change with condition as predicted by the hypothesis, since as the authors point out on p.14 this effect on RT on choice might just reflect non-motivated action or random choice when RTs are fast, but doesn't say anything about the SR vs GD process.

17) Font in Figure 1A in Figure 1E is too small.

18) p.20 – it would be great if the authors could clarify what it might mean that an interaction between RT and condition is observed in Experiment 2 but wasn't in Experiment 1 – why would removing the time interval between cue and stimulus onset cause this change and can the new result be interpreted in light of SR vs GD processes? Or should the GLM suggested above (predicting RTs from choice and condition) also be run here to test that?

19) p.21 – please describe more explicitly how ECG/EMG activity are assumed/hypothesized to relate to SR vs GD

processes.

20) For all three power analyses, please specify the required effect size and statistical test used.

21) p.26 – can the authors give more details about what t being the sample point means in terms of actual timing? Since the iHR and EMG data have much more frequent sampling than the predictors (condition, response, etc, happening only once per trial), what sample is actually used here? Were iHR/EMG values downsampled to one per trial, or were task-related variables upsampled to meet the sampling rate of the physiological data?

22) p.26 – how were the durations of 100 ms for ECG and 20 ms for EMG determined for the cluster-based permutation approach?

23) p.27 – “Heart Rate (HR) values were then z-scored within participants.” – what is the rationale for z-scoring within individuals? I understand this may help take care of individual variability in heart rate, however, what I find problematic is that an individual who exhibit very little variation in heart rate from trial-to-trial will now have values of HR in this regression analysis that are scaled the same as an individual who may show high variability, which renders the interpretation of the resulting coefficients difficult.

24) p.27 – What random effects were included in this model – only random intercepts or maximal random structure as suggested by Barr (2013)? The fact that all interactions (including the 4-way interaction) was included in the model makes me think that the maximal random structure wasn't used. If that's the case, I would consider simplifying the model to be able to include all random effects as well as make the findings interpretable.

25) Why were only ECG-signals hypothesized to be related to avoidance decisions but not EMG signals?

26) Some of the references to the supplemental materials are wrong (for example p.30 of the main text refers to Table D8 for the four-way interaction, when these findings seem to be shown in Table D10). Please cross-check all references to the supplementary materials.

27) Were heart rate and EMG also collected during the subjective evaluation task?

28) p.35 - I am not sure if this question is possible to answer, but I wonder if k may increase in a larger sample? I am assuming that with N=90, the power of reliably identifying 5 clusters is quite low, but with a larger sample size, the findings may be different? This may be a great point to bring up in the discussion.

29) p.35 – when comparing drift rate between Conditions and Class, is there a way to significantly test for an interaction? Or does the fact that the credible intervals for the condition differences overlap between Class A and Class B suggest that there is no interaction?

30) p.38 – how to interpret the finding that drift rate is much lower in Class A than Class B?

31) p.39 – I am not convinced that the results support a “direct causation of GD processes on social avoidance” – the manipulation of predictability may have caused an increase in social avoidance, but the causal role of GD processes cannot be known for sure, and may not be direct (i.e. other factors could be at play)

32) p.42 – if I understand the interpretation of Class A vs Class B correctly, it seems like people in Class A are more driven by GD processes, while those in Class B are more driven by SR processes. I understand this may be a bit simplistic and more nuance is needed; however given the grounding of the paper in the GD/SR dichotomy, it would be helpful to interpret the two classes in light of these processes (this relates to my major comment #1).

Reviewer #3 (Remarks to the Author):

COMMSPSYCHOL-24-0069-T by Sequestro et al.

Thank you for giving me the opportunity to review the manuscript submitted to Communications Psychology by Sequestro et al. The paper reports on three experiments designed to investigate threat avoidance behaviour in a novel VR environment. Specifically, the aim of the study is to disentangle the contributions of stimulus-response associations and goal-directed processes to the avoidance of threats represented by angry avatar facial expressions. The main manipulation of the experimental design is full control over the avoidance of the threatening stimulus (predictable condition), contrasted with random confrontation with an angry avatar (.50), independent of initial avoidance decisions (unpredictable condition). While Experiments 1 and 2 focused on behavioural outcomes and their modelling, Experiment 3 additionally considered cardiovascular and muscular activity.

The main finding, consistent across the three experiments, is that avoidance rates are higher in predictable than unpredictable trials, with avoidance rates above chance level in the latter condition.

Although I am enthusiastic about the VR approach implemented in the study, as well as the authors' goal of replicating the main finding in the three experiments (all with proven sample size), I have a few major concerns about the study.

1) In line with recent frameworks of approach-avoidance behavior, the authors propose two fundamentally different mechanisms underlying avoidance behavior, i.e., SR and GD processes. What I have missed in the introduction of the paper is a clear assumption as to whether SR and GD operate exclusively or – if not – run in parallel or serially. In my view, such a specification would be necessary to better motivate the rationale of the present study. What concerns me most, is that it remains unclear (from my careful reading of the paper) to what extent SR and/or GD processes are elicited by the different structures of predictable and unpredictable trials. Therefore, the two contrasting hypotheses derived at the end of the introduction seem very unspecific.

2) In general, the study lacks specific hypotheses, especially given the large number of variables collected, including individual trait parameters, RTs included in the avoidance rate models, and physiological parameters. Regarding the latter, for example, there is no scientific justification for measuring EMG activity of the flexor pollicis brevis muscle in addition to the overt responses required by the task decisions (there is only a single sentence on p. 21, which is not clearly linked to the

aims of the paper). Trait variables (obtained from questionnaires) are not included in the models of the data from Experiments 1 and 2, but are then implemented in Experiment 3, again without any directed hypotheses. Similarly, the inclusion of the Subjective Evaluation Task in Experiment 3 appears to be incidental, and it remains unclear why the authors related its scores to some of the variables of interest but not to others. In summary, the paper seems to report a series of exploratory analyses which, unfortunately, are not convincingly motivated.

3) Having gained this impression, I was rather surprised by the final "exploratory analysis" section of the paper. The authors justify these additional analyses with the assumption of potential heterogeneity in avoidance behaviour. I fully agree with the authors that such inter-individual differences are usually ignored in experimental studies and that the cumulative power from the three independent samples of their experiments is sufficient to take them into account. However, it remains unclear on what the authors base this assumption in their study; this would be more comprehensible if they had compared models for the individual experiments, e.g. with versus without 'subjectID' as a predictor.

Taken together, the paper reports interesting - albeit predictable - findings from a novel VR paradigm. However, in my view, the overarching research question regarding the contribution and weighting of GD and SR processes in avoidance decisions remains unanswered, which may be due to the operationalization. The manuscript is rich in data and analyses but simultaneously lacks specific hypotheses. As a result, the general discussion (and conclusion section) is inconclusive and often repetitive.

EDITORIAL POLICIES

We ask that you ensure your manuscript complies with our editorial policies and reporting requirements.

To that end, we require revised manuscripts to be accompanied by two completed items: a reporting summary that collects information on study design and procedure, and an editorial policy checklist that verifies compliance with all required editorial policies

- <https://www.nature.com/documents/nr-reporting-summary.zip>>Nature Research Reporting Summary
- <https://www.nature.com/documents/nr-editorial-policy-checklist.pdf>>Editorial Policy Checklist

All points on the policy checklist must be addressed. Your revised manuscript can only be sent back to the referees if these checklists are completed and uploaded with the revision.

Notes: If you have submitted a Stage 1 Registered Report, Review, Primer, Comment, or Perspective you do not need to submit these forms. If you have already submitted these forms, you may disregard this request.

** Visit Nature Research's author and referees' website at <http://www.nature.com/authors>>www.nature.com/authors for information about policies, services and author benefits**

If you experience problems in linking your ORCID, please contact the <http://platformsupport.nature.com/>>Platform Support Helpdesk.

Version 1:

Decision Letter:

Dear Mr Sequestro,

Thank you for your patience during the peer-review process. Your manuscript titled "Social threat avoidance depends on action-outcome predictability" has now been seen by 1 reviewer, I include their comments at the end of this message. I have also editorially reviewed the manuscript with an eye to the concerns of Reviewers 1 and 3 who were unable to rereview. The reviewer and I both appreciate that many concerns raised by the reviewers have been addressed. We are interested in the possibility of publishing your study in Communications Psychology but would like to consider your responses to a few remaining editorial concerns and assess a revised manuscript before we make a final decision on publication.

We therefore invite you to revise and resubmit your manuscript, along with a point-by-point response. Please highlight all changes in the manuscript text file.

Editorially, we think that one central aspect remains to be addressed. The rationale for interpreting threat avoidance in terms of goal-directed processes is that in the unpredictable condition, threat is less avoided than in the predictable condition. Reviewer #1 highlights that with longer response time, goal-directed processes can typically override stimulus-driven responses. This is potentially problematic because it opens the findings to an alternative explanation, which is that in the unpredictable condition, participants deploy goal-directed strategies. In the unpredictable condition, the avatars leave the elevator, and new ones appear, so there is a concrete possibility that participants explore (through goal-directed mechanisms) the impact of their choices on the avatar change (a stochastic event that does not occur in the predictable condition). This possibility has partially been addressed through the analysis aiming at testing learning processes, addressing similar concerns raised by Reviewer #1 and Reviewer #2. However, since in the unpredictable condition, nothing can be concretely learned, the supplementary analyses cannot exclude that the reduction in threat avoidance is due to goal-directed exploration strategies. This is a limitation that warrants some caution in the interpretation of the results and must be addressed in the discussion.

Please, when revising the manuscript, avoid referring to the unpredictable condition as "contingency degradation," which implies that the contingencies acquired in a first phase undergo degradation in a second phase. In the present set of experiments, the association between an action and an outcome is random but does not undergo degradation.

Finally, in the analysis including the latent group, the interaction effects are clearly explained, but it appears that aside from the interactions, there is also a strong main effect. It seems that in general, participants classified as "goal-directed" avoid less social threat than participants classified as "stimulus-driven" in the unpredictable but also in the predictable condition. This effect appears to be counterintuitive given the main claim that social threat avoidance is a goal-directed process. Could you please elaborate on this aspect in the revision?

I am attaching an Editorial Requests Table that details critical reporting requirements for the revised manuscript. Please attend to each item and ensure your manuscript is fully compliant. We are requesting that your manuscript aligns with these requirements as this facilitates the evaluation of your manuscript, reducing delays in re-review and potential future acceptance. If your revised manuscript is not aligned with these requests on major issues, such as those concerning statistics, it may be returned to you for further revisions without re-review. Additional information can be found in our style and formatting guide Communications Psychology formatting guide.

Please use the following link to submit your

- revised manuscript,
- point-by-point response to the editorial comments,
- cover letter (as a separate document),
- the Editorial Policy Checklist (see below),
- the Reporting Summary (see below), and
- the completed Editorial Request Table (attached):

Link Redacted

We hope to receive your revised paper within 8 weeks; please let us know if you aren't able to submit it within this time so that we can discuss how best to proceed. If we don't hear from you, and the revision process takes significantly longer, we may close your file. In this event, we will still be happy to reconsider your paper at a later date, provided it still presents a

significant contribution to the literature at that stage.

Best regards,

Eva R. Pool

Eva R. Pool, PhD
Editorial Board Member
Communications Psychology
orcid.org/0000-0001-5929-1007

REVIEWER EXPERTISE:

Reviewer #2 Modeling and social decision-making

REVIEWER REPORTS:

Reviewer #2 (Remarks to the Author):

The authors have appropriately addressed my comments - I am now happy to recommend the manuscript for publication.

EDITORIAL POLICIES

We ask that you ensure your manuscript complies with our editorial policies and reporting requirements.

To that end, we require revised manuscripts to be accompanied by two completed items: a reporting summary that collects information on study design and procedure, and an editorial policy checklist that verifies compliance with all required editorial policies.

- <https://www.nature.com/documents/nr-reporting-summary.zip>>Nature Research Reporting Summary
- <https://www.nature.com/documents/nr-editorial-policy-checklist.pdf>>Editorial Policy Checklist

All points on the policy checklist must be addressed. Your revised manuscript can only be sent back to the referees if these checklists are completed and uploaded with the revision.

Notes: If you have submitted a Stage 1 Registered Report, Review, Primer, Comment, or Perspective you do not need to submit these forms. If you have already submitted these forms, you may disregard this request.

** Visit Nature Research's author and referees' website at <http://www.nature.com/authors>>www.nature.com/authors for information about policies, services and author benefits**

Communications Psychology is committed to improving transparency in authorship. As part of our efforts in this direction, we are now requesting that all authors identified as 'corresponding author' create and link their Open Researcher and Contributor Identifier (ORCID) with their account on the Manuscript Tracking System prior to acceptance. ORCID helps the scientific community achieve unambiguous attribution of all scholarly contributions. You can create and link your ORCID from the home page of the Manuscript Tracking System by clicking on 'Modify my Springer Nature account' and following the

instructions in the link below. Please also inform all co-authors that they can add their ORCID to their accounts and that they must do so prior to acceptance.

Version 2:

Decision Letter:

Dear Mr Sequestro,

Your manuscript titled "Social threat avoidance depends on action-outcome predictability" has now been editorially reviewed, and I am delighted to say that we are happy, in principle, to publish a suitably revised version in Communications Psychology.

We therefore invite you to revise your paper one last time to address the remaining concerns of our reviewers and a list of editorial requests. At the same time we ask that you edit your manuscript to comply with our format requirements and to maximise the accessibility and therefore the impact of your work.

EDITORIAL REQUESTS:

SUBMISSION INFORMATION:

OPEN ACCESS:

*** TRANSPARENT PEER REVIEW:** Communications Psychology uses a transparent peer review system. On author request, confidential information and data can be removed from the published reviewer reports and rebuttal letters prior to publication. If you are concerned about the release of confidential data, please let us know specifically what information you would like to have removed. Please note that we cannot incorporate redactions for any other reasons.

*** CODE AVAILABILITY:** All Communications Psychology manuscripts must include a section titled "Code Availability" at the end of the methods section. We require that the custom analysis code supporting your conclusions is made available in a publicly accessible repository at this stage; please choose a repository that generates a digital object identifier (DOI) for the code; the link to the repository and the DOI must be included in the Code Availability statement. Publication as Supplementary Information will not suffice.

*** DATA AVAILABILITY:**

Link Redacted

Best regards,

Jennifer Bellingtier

Jennifer Bellingtier, PhD
Senior Editor
Communications Psychology

Eva R. Pool, PhD
Editorial Board Member
Communications Psychology
orcid.org/0000-0001-5929-1007

Response to reviewers

Manuscript Number: COMMSPSYCHOL-24-0069A

“Social threat avoidance depends on action-outcome predictability”

We would like to thank the reviewers for their thorough and helpful feedback on our manuscript. Please find below a point-to-point response to all reviewers’ commentaries. All changes made based on the comments have been tracked in blue in the manuscript.

Reviewer #1:

The authors conducted three studies to investigate the influence of stimulus-response (SR) and goal-directed processes (GD) in social threat avoidance. In an elegant paradigm, participants are faced with a binary choice between two elevators that each have an avatar inside. On threat trials, one of the avatars has an angry face and the other a neutral face. Importantly, a cue indicates to participants if their choice leads to a predictable or unpredictable outcome. In the predictable condition, avoiding or approaching the angry avatar leads to this outcome, whereas this is stochastically determined in the unpredictable condition. Results indicate that participants avoided angry avatars more when the outcome of their choice was predictable, which could be explained by more efficient evidence accumulation in that condition. In addition, physiological data showed that value integration was facilitated by greater cardiac deceleration at the time of choice in the predictable context. Another interesting finding is that participants could be clustered into two groups, one which avoided angry avatars only in the predictable condition, and another which avoided angry avatars in both conditions.

This is a nice series of studies with interesting results for various areas and a well-written manuscript. The analysis was not preregistered, but the predictions follow from the theory. Data and code are available. Below are some of my comments:

C1: The authors hypothesized “if social threat avoidance is driven by SR processes, then there should be no difference in avoidance rates between conditions (as SR processes are insensitive to the predicted outcome).” However, recent work shows that SR processes may only determine behavior under strict time constraints and are quickly controlled by GD processes (Hardwick et al., 2019). While the authors took into account RTs, observed RTs may be different from response preparation time manipulation. I’m not sure if the hypothesis is applicable in the current study design as participants have up to 1.5 seconds to respond. Thus, while the authors have convincingly shown that social threat avoidance is strongly influenced by GD processes, I wonder to what extent the current experiments can speak to initial SR tendencies. In support of this, GD avoidance in the predictable increased with more RTs.

R1: We thank the reviewer for this comment concerning whether our experimental paradigm allows us to derive conclusions about SR tendencies in the face of threatening expressions, which has been raised

as a concern also from the other reviewers and the Editor. Please find below our line of reasoning, which better clarifies why we think that we can conclude that SR responses to this kind of expressions play a negligible role in everyday social interaction, for most but not all individuals.

First, we agree with the reviewer that most of the time “SR processes may only determine behavior under strict time constraints and are quickly controlled by GD processes”, as nicely demonstrated by Hardwick and colleagues (2019). Notably, this paper showed that, after habit formation on the association between four stimuli and correct responses, subjects are more prone to committing habitual errors between 300-600ms, when confronted to a revised stimulus-response-outcome mapping, but not after 600ms due to goal-directed processes overcoming the initial tendency. However, the key difference with our task, is that in the unpredictable condition, no efficient goal-directed process relative to the location of the emotional expression of the avatar could take place, precisely because the response-outcome contingency was broken (contingency degradation; Balleine & Dickinson, 1998). As we state in the introduction, “in uncontrollable contexts, when the contingency between actions and consequences is degraded, the additional computational complexity of GD processes is unnecessary, and SR associations are a more parsimonious policy (Dorfman & Gershman, 2019; Gershman et al., 2021).” (page 4, lines 88-91).

The next question is whether the initial SR tendency, which consist in a “preexisting association between the representation of a specific stimulus and the representation of a specific action” (Moors 2017, p. 2), when not overcome by a goal-directed process, is systematically translated into overt action and thus whether it would impact behavior in our task in the unpredictable condition. Our hypothesis was that, if avoidance is mostly driven by GD processes, and SR associations do not play a prominent role in everyday social decisions, avoidance rates would be at chance in the unpredictable condition. This could either derive from the fact that SR in response to threatening expressions are simply not elicited, or elicited but not translated into overt action even when goal-directed processes are not an efficient policy. Yet, our results proved the “fully GD” account partially wrong, as in a third of our sample, avoidance is high and equal between unpredictable and predictable conditions, suggesting that, at least for some individuals, SR avoidance tendencies are indeed powerful drive of social behavior. Furthermore, despite them being completely unrelated to the ultimate outcome of the action, thus being by definition non-goal-directed (Balleine & Dickinson, 1998), these avoidance responses still correlated with response times. In other words, the longer subjects took to respond, the stronger these SR associations impacted on behavior. This nicely suggests that the early time-window for habitual errors found by Hardwick et al. (2019) might not mean that SR are short-lived, but rather reflect the time needed for GD process to win the competition and overcome the SR association.

This said, we agree with the reviewer that it will be interesting in future studies to manipulate the time available to respond, to test whether more individuals would rely on SR associations under time pressure. Of note, this manipulation could also be pertinent for classical forced-choice approach-avoidance tasks (e.g., AAT), on which a whole lot of the literature on SR tendencies in the face of emotional expressions has been built (e.g., Marsh et al., 2005, 2015; for a review see Kaldewaij et al., 2017), which typically leave up to 2 seconds to participants to respond.

Please find our extensive modifications relative to all these points in the Discussion (page 39, lines 1178-1199).

C2: Related to the first comment, participants may develop goal-directed strategies in this experiment. To this end, it could be interesting to compare the first block to the last block, to see if avoidance is more common in the first block when goal-directed strategies to selectively avoid the angry avatars in the predictable condition may not be as strong.

R2: Thank you for this suggestion. We ran a model including the block (First vs. Last), the condition, the class and the RTs, and found a marginally significant 4-way interaction ($p = .054$, see *Table R2.1*). To better understand the interaction, we ran the same model, separately for each class, and found that the interaction between block and condition was significant only for the GD class (SR: $\beta = -.074$, 95%CrI=[-.693, .554], $p_{>0} = .408$, $BF_{01} = 7.561$; GD: $\beta = .397$, 95%CrI=[.007, .788], $p_{<0} = .023$, $BF_{10} = .567$), suggesting that in the GD class, but not in the SR, avoidance rates increased from the first to the last block only for the predictable, but not the unpredictable, condition (see *Figure 4C*, main manuscript, and *Table R2.2* below). This indicates the GD strategy builds up with time, thanks to participants learning that they can predict the outcome of their action only in the predictable condition. Consistently, this is not true for the group of participants whose avoidance is SR-driven. Notably, this disconfirms the prediction that avoidance is more common in the first block, as suggested by the Reviewer. Also, this contradicts the idea that the “reduction of social threat avoidance in the uncertain condition could derive from goal-directed strategies aiming at avoiding the unpredictable threat (e.g., entering in the elevator with the angry avatar, hoping that that it will switch)”, as suggested by the Editor, as in this case avoidance should decrease over time, which is not the case.

Table R2.1: Regression table for the $p(\text{avoid}) \sim \text{block} * \text{condition} * \text{class} * \text{RTs}$ model

	Est.	Est.Err.	95%CrI-low	95%CrI-up	$p_{>0}$	$p_{<0}$	BF_{01}	BF_{10}	R-hat	ESS
Inter.	.679	.139	.408	.954	1	0	0	7.278e+13	1	10006
Cond.	.105	.205	-.296	.510	.699	.301	10.569	.095	1	9662
Block	.397	.235	-.058	.868	.956	.044	2.478	.404	1.001	9552
RT	1.335	.422	.523	2.171	1	0	.028	36.330	1	11358
Class	-.613	.163	-.931	-.296	0	1	.005	207.391	1	9918
Cond:Block	-.076	.311	-.685	.533	.404	.596	7.620	.131	1	9283
Cond:RT	.532	.620	-.694	1.754	.807	.193	2.857	.350	1	11298
Block:RT	1.286	.719	-.106	2.696	.963	.037	.752	1.330	1	11919
Cond:Class	.022	.242	-.456	.494	.541	.459	10.154	.098	1	9662
Block:Class	-.371	.273	-.912	.169	.086	.914	3.797	.263	1	9726
RT:Class	-1.482	.485	-2.439	-.540	.001	.999	0.042	23.925	1	11418
Cond:Block:RT	-1.487	.927	-3.310	.334	.055	.945	.736	1.359	1	11991
Cond:Block:Class	.464	.367	-.251	1.181	.898	.102	3.070	.326	1	9458
Cond:RT:Class	.777	.721	-.629	2.192	.859	.141	1.953	.512	1	11827

Block:RT:Class	-.613	.815	-2.214	.967	.224	.776	2.353	.425	1	11619
Cond:Block:RT:Class	2.046	1.064	-.039	4.151	.973	.027	.381	2.627	1	12376

Note: Inter: Intercept. Cond: Effect of the Predictable condition on the Unpredictable condition. Block: Effect of the last block on the first one. RT: Response Time. Class: effect of the GD class on the SR class. Est.: Estimate. Est.Err.: Estimate Error. 95%CrI-low and -up: 95% Credible Interval, lower and upper boundary. $P_{>0}$ and $p_{<0}$: proportion of posterior samples greater and lower than zero. BF_{01} and BF_{10} : Bayes Factor for the null (the parameter is not different from zero) and alternative (the parameter is different from zero) hypothesis. R-hat: potential scale reduction factor. ESS: Effective Sample Size.

Table R2.2: Proportion of avoidance responses by class, condition and block (first vs last)

Block	SR class		GD class	
	Predictable	Unpredictable	Predictable	Unpredictable
first	.697	.680	.533	.521
last	.759	.774	.599	.523

We added these analyses to the Supplementary Material (*Supplementary Material L – Avoidance rate across experimental blocks*; pages 43-44).

C3. To determine if avoidance is selected above chance level, the authors report the intercept. However, in particular in experiments 2 and 3, the figures (2B and 3B), it looks like the proportion of choosing avoidance in the unpredictable condition may not be different from chance (.5). I think one sample t-tests comparing the proportion to avoid against chance would perhaps be meaningful. On that note, it would be helpful if the authors reported means for the results.

R3: We are now reporting proportion of avoidance by condition in the manuscript for all experiments, as well as Wilcoxon tests against chance (.5) for the unpredictable condition (page 13, lines 351-357; page 17, lines 498-504; page 27, lines 803-807; page 33, lines 1000-1010). We chose to run Wilcoxon tests instead of t-test as the distribution of avoidance proportions in the unpredictable condition did not always meet the assumption of normality, across experiments (anyway, the results do not change when using t-tests). The results of these tests confirmed that avoidance is greater than chance in the unpredictable condition across the experiments, except for the GD latent class (formerly class A).

C4. The authors state “To test whether a behavior is GD, a classical approach consists in manipulating the outcome value of a response (e.g., by outcome devaluation or contingency degradation; Balleine & Dickinson, 1998).” Strictly speaking, the contingency degradation does not manipulate the outcome value but instead response-outcome contingency.

R4: We corrected the statement (page 4, lines 80-83). Now the text reads: “To test whether GD or SR processes, or a combination of both, are involved in determining behavior, a classical approach consists in manipulating the outcome value of a response (e.g., by outcome devaluation³⁷) or the action-outcome contingency (e.g., by contingency degradation³⁷).”

C5: Moreover, they state “Indeed, GD behavior, but not SR, is sensitive to the prediction of the value of the outcome of the chosen action, as well as to the predictability of the association between actions and outcomes (Balleine & Dickinson, 1998; De Houwer et al., 2023).”. I believe that the term “contingency” would be more precise than “predictable”. In a contingency degradation paradigm, the reinforcer may be predictable but responding is not instrumental. I think the point is that in the “unpredictable” condition responses are “not instrumental”.

R5: We agree and corrected the manuscript accordingly (page 4, lines 83-84), which now reads: “Indeed, while GD behavior is sensitive to outcome values and action-outcome contingencies, SR behavior is not affected by learning or environmental feedback^{37,38}.” We agree that in the “unpredictable” condition responses relative to the threatening avatar’s position cannot be instrumental, as we have now better explained throughout the text.

C6: Finally, they state “Specifically, when an action reliably leads to the same consequence (controllable action-outcome context), GD processes should be privileged as they are more optimal.” However, some work indicates the opposite, where SR are stronger when reinforcers are predictable because less attention needs to be paid in a stable context (Bouton, 2021). Here again, this may be due to the word predictable rather than instrumental.

R6: We see the reviewer’s point, but do not fully agree with it. In details, Bouton and colleagues’ work suggests that: (1) a habit needs a stable contingency between the action and the outcome in order to form and to be maintained as an efficient strategy (less attention); (2) surprising reinforcers which differ from the established contingency can provoke a switch from habit back to goal-directed control (Bouton, 2021). In our task, there is no stable context, as predictable and unpredictable trials are randomly presented, and participants need to pay attention to the cue. Therefore, in our case, no habit can be formed. Instead, we test whether pre-existing habits, notably learned associations between threatening expressions and avoidance, influence behavior. This said, we now better clarified in the introduction that, as in the predictable condition both SR and GD strategies would lead to the same outcome (i.e., threat avoidance), and thus cannot be dissociated, we implemented an unpredictable condition which enables a clear dissociation between the two processes.

In this paper, the authors investigate the balance between stimulus-response associations (SR) and goal-directed processes (GD) during an approach-avoidance task in order to shed light on the computational mechanisms by which people learn and respond to social threat. Approach-avoidance was tested in a virtual reality social context where participants had to decide between entering an elevator with a neutral-face avatar or an elevator with an angry-face avatar. In one condition, the avatars initially present in the elevator were independent from the avatars that would ultimately walk in at the end of the trial (unpredictable) while in another condition the same avatars would stay on from the beginning (predictable). The authors found that participants were generally likely to avoid the elevator with the angry avatar, and that this tendency increased in the predictable condition, consistent with an increased reliance on GD processes when the environment is predictable. This finding was also accompanied by changes in drift-diffusion model (DDM) parameters, particularly the drift rate, as well as changes in heart rate deceleration, and was found to vary across individuals.

I generally really enjoyed reading this paper and the progression between the three experiments. I do believe that it makes a novel and substantial contribution to the topic of social avoidance behavior. However, I do have quite a lot of comments, including a few major concerns, that I'd like to see addressed before recommending the manuscript for publication.

Major comments:

C7: The study rationale is mainly grounded in the SR vs GD dichotomy, but the specific analysis methods and interpretation of the findings often fails to make the link with these processes. A few examples:

C7a: I understand that SR is assumed to be an automatic response and therefore should not be influenced by the predictability of outcomes – however, seeing a change in behavior when increasing predictability does not mean that SR is not driving behavior. SR could be driving behavior together with GD, and the 'level' of SR remains the same when predictability increases. It seems key to obtain an independent behavioral measure of SR and/or manipulate SR in an independent way from GD.

R7a: We thank the Reviewer for giving us the opportunity to better clarify our hypothesis. We fully agree that both SR and GD might participate in the predictable condition. This is why the avoidance rates at the unpredictable condition are key to disambiguate between SR and GD. We now better clarify this aspect in the Introduction (page 5, lines 107-119), now reading:

“Overall, we expected that participants would avoid the angry avatar more than chance in the predictable condition as, irrespective of the type of process involved, both GD and SR processes should lead to threat avoidance. If GD processes are involved in the decision, then avoidance rates should decrease in the unpredictable context, because of contingency degradation. On the contrary, such reduction should not be observed if avoidance is solely driven by SR associations. Importantly, a simple

reduction in avoidance rate under unpredictability would not imply that SR processes are not involved in the decision, as SR processes might just keep operating in the unpredictable context as they do in the predictable one. Conversely, an avoidance rate at chance level in this condition would suggest that SR associations are not driving avoidance, either because SR action tendencies in response to threatening expressions are simply not elicited, or because they are not overtly influencing avoidance. Therefore, by comparing the avoidance patterns between the two conditions, we tested the effect of GD and SR processes over social-threat avoidance.”

The results from the three experiments showed that avoidance was greater in the predictable vs. unpredictable condition, but still above-chance in the unpredictable. This signifies that, despite contingency degradation decreasing avoidance rates, SR mechanisms are also at play. Therefore, we wondered whether this mixed result was the effect of both strategies being present within participants, or across them. The class analysis clarified that while the majority of participants used a GD strategy (i.e., avoidance at chance in the unpredictable condition and above chance in the predictable one), one third of them used a SR strategy (avoidance rates equal between conditions and above chance). To better clarify this, we renamed the former A and B classes, “GD” and “SR” classes, respectively, as suggested by Reviewer 2 (see C38).

Concerning the reviewer’s last point, it would be very interesting to orthogonally manipulate GD and SR processes. Aside from manipulating the time constraint for the response (see R1), another idea would be to manipulate the perceived distance to threat, following the predatory-imminence model (Fanselow, 1988; Mobbs, 2020), which suggests that imminent threats are more likely to provoke SR-driven reactions. For instance, a condition could be added to the task, in which when entering the elevator with the angry individual, the subjects get aggressively approached and attacked in VR. Some studies in VR have recently employed such manipulation (e.g., Gado and Gamer, 2024; Lu et al., 2023), which could be orthogonalized with the predictability of the outcome and provide a more exhaustive picture of mechanisms involved in social threat avoidance. We added this to the discussion (page 39-40, lines 1200-1211).

C7b: Related to the above, the DDM makes a clear prediction that changes in drift rates are reflective of changes in GD processes, but nothing in the model seems to be able to independently quantify or characterize SR processes. My understanding of DDMs is that they also usually include a starting point parameter – which in this case seems like it would best capture the biased preference to avoid the angry avatar (and could also constitute the signature of SR processes?)

R7b: Thank you for the opportunity of explaining why we did not estimate the starting point. The contingency between the response (left, right) and the outcome (approach, avoidance) was not known a priori to the subjects, as it depended on the information provided by the stimulus (i.e., the position of the angry avatar). Under these conditions, the typical recommendation is not to fit the starting bias (e.g., Fontanesi et al., 2019; Perters and D’Esposito, 2020; Urai and Donner, 2022). It is true that in some of our previous studies we did estimate the starting point bias in a similar task, arguing that fast sub-cortical automated reactions to threat could in principle act on the starting point parameter (i.e.,

influence choice even when RTs were very fast), similarly to what a cue preceding the start of the evidence accumulation would do (Mennella et al., 2020; Grèzes et al., 2021). Nonetheless, in both papers we did not find a modulation of emotional expressions on the starting point either in line with the fact that this rapid influence does not exist in the task or, possibly, that the starting point is not a well-suited parameter to capture this, under these conditions. Due to this ambiguity in the interpretation, we prefer to stick to recommendations in the present paper, also because we think that the intercept of the GLM models is a much straightforward indicator of the presence of SR, as the response to the next point will hopefully clarify. In our opinion, the DDM still provides very useful information concerning the manipulation of evidence accumulation via outcome (un)predictability.

C7c: In the GLM on avoidance decisions – could the intercept be interpreted as a signature of SR processes? If so, could the authors find a way to actually assess the relative contribution of SR vs GD processes in the two conditions (maybe by comparing the standardized intercept with the standardized effect of condition) so their hypothesis can be more directly tested?

R7c: The intercept of the model represents the avoidance rate in the Unpredictable condition (as the condition variable is treatment coded with Unpredictable = 0 and Predictable = 1). Therefore, we agree that a statistically significant intercept signaling higher than chance avoidance rate in this condition is a signature of SR processes impact on decision. Furthermore, the effect of condition represents the log-odds change in the probability to avoid/approach, induced by outcome predictability, and can indeed be interpreted as a signature of goal-directed process.

As suggested, we performed a 2 (effect: intercept vs. condition) by 2 (class: SR vs. GD) mixed ANOVA with the standardized estimate of the coefficients as a dependent variable. We found a significant effect by class interaction ($F_{(1,176)}=295.968$, $p < .001$, $\eta_g^2=.627$), showing that the SR class presents a greater standardized intercept compared to the condition effect ($emm=-2.24$, $p_{bonf}<.001$), while the opposite is true for the GD class ($emm=1.06$, $p_{bonf}<.001$; see the Figure below). Following the reasoning that the intercept is a trace of SR processes and that the condition effect is a trace GD processes, we can conclude that avoidance is mostly driven by SR processes in the SR class and GD processes in the GD class (in this line, the effect of condition in the SR class is not different from the estimate of intercept in the GD class ($emm=.155$, $p_{bonf}=1$, $p_{uncorrected}=.253$)). Finally, the estimate of intercept in the SR class is greater than the estimate of the condition effect in the GD class ($emm=-1.018$, $p_{bonf}<.001$), explaining the greater avoidance rate in the SR class. This confirms the results of the odds-ratios of the two coefficients (*Supplementary Material D – Tables D23 and D25*), which reveal an intercept non-credibly different from 0 and a condition effect credibly above 0 in the GD class, and the opposite pattern in the SR class.

We added these analyses in Supplementary Material (*SM L – Test on GLMMs standardized coefficients, pages 41-42*).

Figure R7c.1: Group level standardized estimates and relative 95%CrI for the Intercept and Condition effects.

C8: I have a bit of a general concern with the way avoidance is implemented in the main task and the subjective evaluation task. I understand that participants are either choosing the neutral over the angry avatar, or that they are rating how much they'd like to enter the elevator with the neutral avatar, but both of these behavioral metrics would be more directly described as approaching the neutral avatar (rather than avoiding the angry avatar). Could the findings be different in any way if "true" avoidance had been measured (i.e. the tendency to not approach at all)? This would be worth discussing.

R8: We thank you for the suggestion and have added this paragraph to the Discussion (page 40, lines 1212-1234).

"[...] it is worth noting that behavior in the current task can be described both in terms of avoiding the angry avatar or approaching the neutral one. It is a matter of debate whether avoidance is driven by appetitive (i.e., reaching safety) or aversive (i.e., avoiding threat) motivation, or both¹¹⁶. Indeed, approaching safety and avoiding threat are necessarily intertwined ("if safety is the absence of danger, avoidance is a source of safety"¹¹⁶), but at the same time appetitive and aversive motivation might be partially dissociable in terms of neural correlates. Indeed, behavior driven by positive vs. negative reinforcers is in part sustained by common brain networks^{117,118}, but the overlap is not complete, as some regions of the brain seem to respond selectively to each kind of reinforcer^{48,117} and different neurotransmitters may be implicated (for a review, see¹¹⁹). Here we do not have further arguments to dissociate between the two possibilities. Nonetheless, it is important to note that the effect of the predictability of the outcome on the cardiac response is conditioned to the presence in the scene of the angry avatar (i.e., no difference between conditions for neutral trials). This suggests that, at least physiologically, the decision process when approaching neutral is not the same when a threat is present or absent in the scene. To study threat avoidance without the confound of safety approach, it would be of interest to replicate this study introducing trials in which participants can refrain from entering

any elevator, or possibly trials in which they can enter an empty elevator. It is noteworthy that, in such a scenario, depending on each individual's social motivation, participants might always prefer to avoid (or to approach) the elevator with the avatar, irrespective of its emotional expression. This would move the research focus from social threat avoidance to general social avoidance. Nonetheless, due to the relevance of general social avoidance for several psychopathological conditions, from social anxiety to depression, the topic merits further investigation."

C9: I understand that most of the model-free and modelling analyses focus on Threat trials as they aim to analyze the tendency to approach vs avoid the angry avatar. That said, what was the rationale for having neutral trials in the task if those were not considered in any of the analyses? I assumed the goal of the neutral trials may have been to control for any effect of the identity of the avatar beyond the facial expression, which would be worth determining. Given that EMG amplitude was found to differ between threat and neutral trials, it would be great if there was a way to assess any differences in behavior as well (e.g. with regards to RTs, subjective valuation, and/or influence of avatar identity).

R9: The rationale for having Neutral trials was mainly to dissimulate the presence of threatening expressions in the task. As reported in the introduction (page 4, lines 99-100) "Importantly, the presence of different facial expressions of the avatars was never explicitly mentioned to participants, in order to investigate spontaneous avoidance behavior." Furthermore, this design allowed us to test differences between neutral and threat trials in terms of response times and physiological responses. To render this clearer, we now added to the "*Experiment 1 – Materials and methods – VR task*" section (page 7, lines 187-191):

"While most analyses were focused on participants' behavior in Threat trials, the addition of Neutral trials was motivated by the goal of dissimulating as much as possible the presence of threatening expression. Furthermore, this allowed us to run control analyses by testing differences between neutral and threat trials in terms of response times and physiological responses (see Experiment 3)."

Results on cardiac activity for the neutral trials was already presented in the original version of the paper and helped strengthening the conclusion that stronger cardiac deceleration in the predictable vs. unpredictable condition is specific to threat trials (*Supplementary Material I, Figure 15*). We now added the results of models fitting response times by condition and presence of threat in the trial (i.e., $RT \sim \text{condition} * \text{threat}$) in the supplementary material (*Supplementary Material L – Response Times*, page 38) and found no significant effects of threat nor condition. Finally, the subjective evaluation task only presented threat trials to reduce experiment time.

Concerning the effect of each identity on the main results, since avatars were always presented in dyads, we focused on the effect of individual dyads of avatars. To this aim, we ran additional models with the random effect of (SubjectID) + (Dyad) (see *Supplementary Material L – Effect of individual dyads*, page 40-41). Overall, results did not change, showing generalizability beyond the specific avatars employed.

C10: The motivation for the specific analyses run with the self-report questionnaires is unclear – it would be great if the authors could better motivate the rationale for these analyses in the introduction with appropriate literature reviews on what symptom dimensions is expected to be associated with signatures of GD versus SR processes, thus justifying the subsequent choices of associations being tested. More specifically, what is the rationale for only examining correlations between self-report measures and the condition*RT interaction? Why were pairwise correlations run for this latter analysis but a repeated measures ANOVA for the data from the Subjective Evaluation task? Was correction for multiple comparisons performed (in the case of running the same analysis multiple times with different questionnaires)? And finally, was the multi-collinearity between questionnaires established as well as specificity of the findings? For example, for the findings about the effect of state and trait anxiety on subjective evaluation of approach vs avoidance, was a single ANOVA run with both state and trait anxiety as predictors (as well as the other scales), or were two separate ANOVAs run? In the first case, this may be problematic given that state and trait anxiety are generally too highly correlated in the population, which would lead to effects in opposite directions (as reported) given that the amount of unique variance is too small to be properly assigned to each measure. In the second case, correction for multiple comparisons should be run as well as ensuring that the effects are robust to controlling for other scales. My recommendations for these analyses would be to first test the association with each self-report with an a priori hypothesized association, correcting for multiple comparisons, then ensure that any significant association is robust to controlling for other self-report measures, other behavioral metrics from the task, as well as demographic information (e.g. age and gender, if available). Alternatively, an option could be to perform some dimensionality reduction on the questionnaire scores (or subscales or items, though for the latter the sample size may be too low) in order to extract symptom dimensions that are less correlated with each other.

R10: Thank you for this comment, we agree with your remarks. What motivated the inclusion of these measures in the first place is that numerous studies have shown that social interactions and decisions can be affected by depression (Grèzes et al., 2023), changes in positive affect (Grèzes et al., 2021), anxiety and social anxiety (Rum et al., 2023). Nonetheless, our main question is related to the relative contribution of SR and GD processes to social avoidance decisions, and, to our knowledge, no unique association between specific affective traits (e.g., depression, anxiety) and one of the two processes can be clearly anticipated. Furthermore, we studied a non-clinical sample, in which these scores vary limitedly. For these reasons, as well as for the fact that we already report extensive analyses on behavioral and physiological measures, across a series of three studies, as well as the final class analysis, we think that the pertinence of the influence of the affective traits on the results is of secondary importance. Therefore, we decided not to include the analyses related to the questionnaires in the revised version of the paper. Nonetheless, due to the potential pertinence, especially in clinical samples of these measures, we keep descriptive tables for each experiment's sample in the supplementary material as we did in the previous version (*Supplementary Material, Table G2*).

We therefore rewrote the “*Experiment 1 – Materials and Methods – Self-Report Measures*” section (pages 9-10, lines 245-261) as follows:

“Numerous studies have shown that social interactions and decision-making can be affected by changes in positive affect⁴⁵, depression⁴⁶, anxiety and social anxiety⁴⁷, as well as reward/punishment sensitivity⁴⁸. Therefore, participants completed a series of self-administered questionnaires to account for these variables in our sample. Of note, characterizing the effect of these affective and personality traits on behavior is beyond the scope of the present article, which focuses on the balance between SR and GD processes in social threat avoidance. We nonetheless include descriptive results from self-report measures including the observed Cronbach alphas in *Supplementary Material G - Table G2*.

The administered questionnaires were:

- The Positive and Negative Affect Schedule (PANAS⁴⁹), to measure affect;
- The Patient Health Questionnaire depression scale (PHQ-8⁵⁰) to measure depressive symptoms;
- the State Trait Anxiety Scales (STAI⁵¹) and the Liebowitz Social Anxiety Scale (LSAS⁵²), to measure anxiety and social anxiety;
- The Behavioral Inhibition/Behavioral Activation scales (BIS-BAS⁵³).

Finally, to track possible cybersickness symptoms due to VR, participants filled in the Simulator Sickness Questionnaire⁵⁴.”

C11: I find it worrying that there is no main effect of SV on p(avoid angry) in Table D10 or from visually examining Figure 3I. This suggests that people’s subjective valuation of the angry vs neutral avatar did not overall influence their choices. Maybe this main effect does not come out in the model in Table D10 because of the model complexity? It would be interesting to test if this main effect is there in a simple model predicting choice from SV only.

R11: Indeed, the SV difference (avoidance-approach) alone does not predict avoidance rates ($\beta=.123$; 95%CrI=[-.210, .475]; $BF_{01}=11.597$). We disagree with the reviewer that this is worrying, as it seems to us that this supports the claims that we have made in the paper, namely that:

- 1) Avoidance can be goal-directed, and therefore dependent on subjective value, only in the predictable condition.
- 2) Avoidance choices are goal-directed only when participants take sufficient time to respond (see the “C” plots of each figure which show that the difference between conditions emerges with increasing response times).
- 3) Value integration in choice is correlated with increased cardiac deceleration at the time of choice, in line with previous models (Livermore et al., 2021).

We think, on the contrary, that finding a simple effect, and not a four-way interaction between SV, condition, RT and HR would be worrying, as it would disconfirm one or all the previous claims.

C12: It's a bit hard to understand the rationale for why heart rate change was used in some analyses (e.g. Figure 3F), but then heart rate seems to be the variable used in predicting avoidance rates (Figure 3I). Is it a faster deceleration or a slower heart rate that’s associated with more avoidance in the predictable condition? If deceleration is the key metric, shouldn’t a trial-by-trial index of deceleration (rather than minimum heart rate) be used in the GLM from Figure 3I?

R12: Indeed, all analyses are on heart rate change from baseline (so a measure of deceleration). The only difference is that the analyses on time series involved sample-by-sample analyses (i.e., for the detection of the cluster of effect). We corrected the manuscript and figures to make this clearer and avoid any confusion. We are now referring to the previously called “minimum iHR” as “maximum iHR change from baseline” (page 26, lines 779, 781 and 786; page 28 line 854).

C13: In the partial discussion sentence on p.32-33 (“Cardiac deceleration was also found to facilitate the integration of subjective outcome values into decisions only in predictable trial”), please refrain from using words like “facilitate”, since the causal association between cardiac deceleration and choice was not directly tested. Related to my point above about the distinction between heart rate and deceleration, it would be key to establish that the findings are specific to cardiac deceleration, and not just slow heart rate, in order to make such a claim.

R13: We modified the manuscript accordingly (page 30, lines 909-911), which now reads: “Cardiac deceleration was also found to be associated with a greater integration of subjective outcome values into decisions only in predictable trials, when response times were long enough, consistent with previous studies^{74,76,89}.”

C14: The exploratory analyses of heterogeneity are a great addition to the paper, especially to give more insights into individual differences in DDM parameters (which couldn’t be reliably estimated) and potentially in self-report symptom measures (although there were no association, but maybe this would change depending on how my point #4 above is addressed). My main concern is that by definition, the plots shown in Figure 4C-D, and corresponding stats on p.35 (avoidance rate paragraph) are somewhat circular. While it’s okay to show those illustrations as a sanity check to characterize the behavioral differences between the two clusters and make sure those match the differences from Figure 4B, it’s obvious that a 3-way condition*RT*class interaction will emerge given that the clusters were defined based on the condition*RT model, so I would refrain from any statistical report/inference on these findings.

R14: We do not fully agree that the analysis is circular. Indeed, fitting the Finite Mixture Model on the linear model composed by the intercept, the main effects of RT and condition, and their interaction, could suggest the existence of separate classes of participants that differ on at least one of the previous effects (Grün and Leisch, 2008), and not necessarily on the higher-order interaction. Therefore, it is informative to identify the effect(s) on which the classes differ. In our case, we found a difference between classes on the condition*RT interaction, but also, for example, in the intercept (which emerged as a main effect of class in the combined analysis). Both results were informative, as the three-way interaction suggests that some subjects use a GD strategy and some others a SR one, and the main effect of class adds to this that the SR group has higher avoidance rates overall. Therefore, we would prefer to keep this analysis in the paper, consistently with previous studies adopting the same modelling approach (e.g., Nook et al., 2022).

C15: The low sample size of the experiments should be noted as a limitation, especially with regards to the heterogeneity and individual difference analyses. Effect sizes of associations between cognitive processes and self-report measures are known to be low, however, the studies were not powered with these low effect sizes in mind.

R15: First, we removed the correlations with self-report measures in the revised version of the paper (see R10). Still, we implemented the reviewer's suggestion relatively to the sample heterogeneity, so now the beginning of the "Limitations" section (page 41, lines 1237-1241) reads:

"[...] although we selected the sample size for each experiment using appropriate power simulations for the condition effect, these experiments may be underpowered with respect to other effects. A larger scale replication of the paradigm would ensure greater power, for example, to identify more latent classes of behavior, providing a finer-grained picture of interindividual differences in social avoidance."

Minor comments (in rough order of text):

C16: From the abstract, it is unclear how the improved performance, increase drift rate and greater cardiac deceleration in the predictable condition helps separate SR versus GD tendencies.

R16: We have now rewritten the abstract to make this aspect clearer. The abstract now reads (page 2):

"Recent research suggests that goal-directed processes, in addition to stimulus-response associations, play a central role in social threat avoidance. To test this, we manipulated outcome predictability in avoidance decisions from avatars displaying angry facial expressions by degrading action-outcome contingencies, which affects goal-directed processes. In three virtual reality experiments, participants avoided more when they could predict the outcome of their actions, indicating goal-directed processes. However, an above-chance avoidance rate in the unpredictable condition suggested stimulus-response associations also played a role. We identified two latent classes of participants: the "goal-directed class" showed above-chance avoidance only in the predictable condition, while the "stimulus-response class" showed no difference between conditions but had a higher overall avoidance rate. The goal-directed class exhibited greater cardiac deceleration in the predictable condition, associated with better value integration in decision-making. Computationally, this class had an increased drift rate in the predictable condition, reflecting the increased value estimation of threat avoidance. In contrast, the stimulus-response class showed higher responsiveness to threat, indicated by an increased drift rate for avoidance and increased muscular activity at response time. These results support the central role of goal-directed processes in social threat avoidance and reveal the physiological and computational correlates of this decision."

C17: What were the participants instructed to in terms of what their goal is in this task?

R17: Participants were only instructed to freely choose to enter in the elevator they preferred. They were further informed of the contingency between the cue (the elevator’s arrow) and the fact that the avatars would stay or leave the elevators after participants’ response. No other explicit instructions were given, and the presence of threatening expressions was never mentioned. Overall, we chose this experimental setting in order to investigate spontaneous avoidance behavior. We further stressed this aspect in the “*Experiment 1 - Methods and material – VR task*” section (page 8, lines 218-223).

C18: Was there an effect of the gender of the avatars?

R18: Interestingly, when pooling experiments 2 and 3, there was a three-way interaction between condition, RTs and avatar’s gender ($\beta=-1.020$; 95%CrI=[-1.819, -.233]; $p>0=.006$; $BF_{10}=3.804$) signaling greater avoidance rates with time in the unpredictable condition when facing male vs. female avatars. This might suggest that male avatars are perceived as more threatening, despite this conclusion not being supported by a difference in subjective value (see *Supplementary Material L – Effect of avatars’ sex*, pages 38-40). Of note, adding participants’ sex as a predictor in the model did not improve model fit.

C19: Thank you for sharing videos of demo trials, this is very helpful in understanding the task – however, I am confused as to why the avatars’ faces are blurred in this video? Also, it seems that in the unpredictable trial, when the avatars walk out of the elevator, it is the same avatar that walks back in rather than a new avatar. But maybe it’s the blurred face that gives this impression.

R19: The faces were blurred because of copyright reasons with the Radboud Face Database that we used to create our 3D avatars. However, on a second read of the terms and conditions of the database, we realized that publishing those faces as stimulus example is not a problem, so we now uploaded videos without any blur on OSF. Importantly, the avatar walking in the elevator after choice in the unpredictable condition is never the same as the one present at the moment of choice. We further

stressed in point in the "*Experiment 1 – Materials and methods – VR task*" section (page 8, lines 210-211).

C20: Is it possible that participants are trying to learn something in this task, especially in the unpredictable condition, whereby even though they are told the new avatars could display an angry vs neutral face with a 50% probability, they may still try to learn whether choosing the elevator with the angry or neutral avatar first has more chance of “attracting” the new angry versus neutral avatar. It may be worth running a quick behavioral check, such that looking at the effect of past trial outcome on subsequent trial choice, or plotting behavior across trials, to ensure that there is no learning effects.

R20: Thank you for the suggestion. First, participants were never “told the new avatars could display an angry vs neutral face with a 50% probability”, as emotional expressions were never mentioned, and we now hope that this is clearer. Concerning the Reviewer’s commentary, and in response also to Review 1 (R2), we are now reporting in the manuscripts plots for each experiment representing behavior across experimental blocks. Furthermore, we indeed found that avoidance rate increased in the predictable vs. unpredictable condition between block 1 and 4, only in the GD class. This confirms that the GD strategy builds over time as the subjects learn that threat-related outcomes are predictable only in one of the two conditions (see also *Supplementary Material L – Avoidance rate across experimental blocks*, pages 43-44).

Furthermore, we ran the suggested behavioral check, by modeling the probability in the current Unpredictable trials (t) to repeat the action (avoid or approach) performed in the previous unpredictable trial (t-1). We focused on the pooled sample of experiments 2 and 3 and we added as predictors in the model the outcome in t-1 (i.e., being in the elevator with the neutral or the angry avatar after the avatar switch), the response times (grand average centered), the latent class (SR=0, GD=1), and the interactions across these predictors. We kept a maximal random structure for the within subjects effects in the model. As shown in the regression table below, we found no effect of previous outcomes on the probability of repeating the previous action in unpredictable trials. Were participants trying to learn something (e.g., approaching the angry avatar hoping that it would switch to a neutral one), we should have observed that if the previous approach decisions resulted in a safe outcome (i.e., being near the neutral avatar), then the probability of repeating this action would have increased. This is not the case for any of the classes. Of note, we do find a greater probability of repeating the previous action in the SR class (i.e., a credible model intercept), which is greater with greater RT (credible RT effect). This is just reflecting the increase tendency to avoid in the SR class: if these participants are overall more likely to avoid, then it’s more likely that they repeat this action between t-1 and t. Confirming this interpretation, we also find that this is not the case for the GD class (i.e., credible class effect).

	Est.	Est.Err.	95%CrI-low	95%CrI-up	p>0	p<0	BF ₀₁	BF ₁₀	R-hat	ESS
Inter:	.461	.105	.258	.668	1	0	0	87520.012	1	11805
Out.Safe	.279	.175	-.064	.623	.946	.055	3.957	.253	1	13643
RT	.742	.323	.114	1.380	.989	.011	.559	1.790	1	10513

class	-.432	.126	-.682	-.185	0	1	.108	9.217	1	11341
Out.Safe:RT	-.117	.553	-1.192	.968	.415	.585	4.410	.227	1	14728
Out.Safe:Class	-.238	.209	-.651	.180	.128	.872	6.455	.155	1	13503
RT:Class	-.715	.378	-1.465	.019	.028	.972	1.128	.887	1	11707
Out.Safe:RT:Class	.182	.659	-1.085	1.480	.609	.391	3.706	.270	1	14742

Note: Inter: Intercept. Out.Safe: Safe outcome in the previous trial. RT: Response Time. Class: effect of the GD class on the SR class. Est.: Estimate. Est.Err.: Estimate Error. 95%CrI-low and -up: 95% Credible Interval, lower and upper boundary. $p_{>0}$ and $p_{<0}$: proportion of posterior samples greater and lower than zero. BF_{01} and BF_{10} : Bayes Factor for the null (the parameter is not different from zero) and alternative (the parameter is different from zero) hypothesis. R-hat: potential scale reduction factor. ESS: Effective Sample Size.

C21: p.10: “we also fitted RTs (grand average centered) and their interaction with the Condition as predictors” – this sentence is unclear – saying “we fitted RTs” suggests that you ran a separate GLM to predict RTs from other variables like Condition, but the rest of the sentence suggests that RTs were included as a predictor of choice? I understand from the results section that the latter seems to be what was done, in which case, the sentence in the methods should read “we also added RTs” (rather than fitted).

R21: We corrected the wording in the manuscript accordingly by changing “fitted” with “added” (page 11, line 296).

C22: A GLM actually fitting RTs may be interesting (e.g. something along the lines of $RT \sim \text{avoid_choice} * \text{condition}$) – this would be the correct model to test whether RTs change with condition as predicted by the hypothesis, since as the authors point out on p.14 this effect on RT on choice might just reflect non-motivated action or random choice when RTs are fast, but doesn’t say anything about the SR vs GD process.

R22: We fitted the models on RTs on experiments 2 and 3 pooled, as suggested, reporting and interpreting the results in the supplementary material (see *Supplementary Material L – Response times*; page 38). Of note, we prefer not to include these analyses in the main text as our hypothesis did not concern response times.

C23: Font in Figure 1A in Figure 1E is too small.

R23: We changed the figures and increased the font size accordingly.

C24: p.20 – it would be great if the authors could clarify what it might mean that an interaction between RT and condition is observed in Experiment 2 but wasn’t in Experiment 1 – why would removing the time interval between cue and stimulus onset cause this change and can the new result be interpreted in light of SR vs GD processes? Or should the GLM suggested above (predicting RTs from choice and condition) also be run here to test that?

R24: Thank you for pointing this out. We added this interpretation in the “*Experiment 2 – Partial discussion*” (page 19, lines 553-562) section:

“Interestingly, whereas in Experiment 1, the difference between conditions was independent of RTs, in Experiment 2, we found that this difference was more evident with slower RTs as reflected by a condition by RTs interaction. Note that in Experiment 1, participants only had to accumulate evidence for the location of the threatening/neutral expressions, as they already had the time to process the cue indicating the predictable/unpredictable condition before the stimulus onset. Consequently, even when responding rapidly they may have had time to combine the information about outcome (un)predictability with threat location. Under these circumstances, even the faster RTs on the distribution would be slow enough to allow for the condition effect to manifest itself (see Figure 1C), making the condition by RTs interaction less likely to emerge.”

C25: p.21 – please describe more explicitly how ECG/EMG activity are assumed/hypothesized to relate to SR vs GD processes.

R25: Please see response to reviewer 3 on the same concern (R40).

C26: For all three power analyses, please specify the required effect size and statistical test used.

R26: We now detailed our simulation approach for power calculation in the beginning “*Experiment 1 – Material and method – Participants*” section (pages 5-6, lines 127-136), which reads:

“The simulation approach for power calculation involves fitting a Generalized Linear Mixed Model (GLMM) to an initial dataset (in this case, a binomial GLMM to the pilot dataset) to extract fixed and random effect parameters. These parameters are then used to generate data for a specified number of artificial participants, on which the same GLMM is fitted. This process is repeated 1000 times for each sample size to test. The power is calculated as the proportion of artificial datasets that show a significant effect ($p < .05$) of the desired parameter out of the total 1000 datasets⁴¹. Compared to analytical methods for power calculation, the simulation approach has the advantage of being feasible for GLMMs, and of considering inter-individual variability (i.e., the random structure of the model).”

C27: p.26 – can the authors give more details about what t being the sample point means in terms of actual timing? Since the iHR and EMG data have much more frequent sampling than the predictors (condition, response, etc, happening only once per trial), what sample is actually used here? Were iHR/EMG values downsampled to one per trial, or were task-related variables upsampled to meet the sampling rate of the physiological data?

R27: Through this methodology we fit each time-point of the physiological series (iHR or EMG) separately by one GLM for each point and for each participant. Therefore, t represents 100 ms of the iHR signal (interpolated at 10Hz) and 1ms of the EMG signal (sampled at 1kHz). Following analyses are based on time-cluster permutations of parameters’ time-series estimated in this way.

To make this clearer we rephrased the manuscript (page 25, lines 739-748) in the following way:

“Through this method, individual time points of physiological signals (iHR or EMG amplitude) from each participant were fitted separately trial-by-trial using a Generalized Linear Model (GLM) with the following formulas:

$$iHR_{(t)} \text{ or } EMG_{(t)} \sim \beta_{0(t)} + \beta_{condition(t)} + \beta_{threat(t)} + \beta_{condition*threat(t)} + \beta_{RT}$$

$$iHR_{(t)} \sim \beta_{0(t)} + \beta_{condition(t)} + \beta_{outcome(t)} + \beta_{condition*outcome(t)} + \beta_{RT}$$

$$EMG_{(t)} \sim \beta_{0(t)} + \beta_{condition(t)} + \beta_{response(t)} + \beta_{condition*response(t)} + \beta_{RT}$$

with t being the time-point, representing 100 ms in the case of iHR (interpolated at 10Hz) and 1 ms in the case of EMG (sampled at 1kHz).”

C28: p.26 – how were the durations of 100 ms for ECG and 20 ms for EMG determined for the cluster-based permutation approach?

R28: Concerning the duration values, they were arbitrarily chosen to provide sufficient time resolution, while the difference in duration between the two signals was determined by the higher frequencies of the EMG compared to the iHR changes. Cluster-based permutation methods always involve a somehow arbitrary choice on several parameters (including for instance the initial threshold used to identify significant contrasts), and the only way to assess the robustness of the findings is to compare with other sets of parameters (Maris and Oostenveldt, 2007), which we did, and found no influence on the results.

C29: p.27 – “Heart Rate (HR) values were then z-scored within participants.” – what is the rationale for z-scoring within individuals? I understand this may help take care of individual variability in heart rate, however, what I find problematic is that an individual who exhibit very little variation in heart rate from trial-to-trial will now have values of HR in this regression analysis that are scaled the same as an individual who may show high variability, which renders the interpretation of the resulting coefficients difficult.

R29: What we are interested in here is intra-individual variability. In this sense, results after z-scoring within individuals are in our opinion easily interpretable reflecting how each person’s cardiac activity responds to the stimulation compared to their usual activity. On the contrary, z-scoring between individuals would make subjects that have higher absolute variability count more in the analysis, limiting the generalizability of the findings. Accordingly, this is a common approach in other papers investigating cardiac responses in emotional contexts (e.g., Engelen et al., 2023).

C30: p.27 – What random effects were included in this model – only random intercepts or maximal random structure as suggested by Barr (2013)? The fact that all interactions (including the 4-way

interaction) was included in the model makes me think that the maximal random structure wasn't used. If that's the case, I would consider simplifying the model to be able to include all random effects as well as make the findings interpretable.

R30: The model already contains all random effects, as suggested by Barr (2013). Bayesian Hierarchical modeling allows for keeping this complexity and still reaching a convergence in parameters estimate.

C31: Why were only ECG-signals hypothesized to be related to avoidance decisions but not EMG signals?

R31: Please see R40.

C32: Some of the references to the supplemental materials are wrong (for example p.30 of the main text refers to Table D8 for the four-way interaction, when these findings seem to be shown in Table D10). Please cross-check all references to the supplementary materials.

R32: References to supplementary materials have now been cross-checked and corrected where necessary.

C33: Were heart rate and EMG also collected during the subjective evaluation task?

R33: We did not detach ECG electrodes not to waste time. We instead detached EMG sensors as subjects were responding using a mouse instead of clicking on a controller (electrodes might have interfered with the movement). As planned, we did not analyze ECG signal during this task because of the faster timing compared to the main, which did not allow for cardiac activity to go back to baseline between each trial.

C34: p.35 - I am not sure if this question is possible to answer, but I wonder if k may increase in a larger sample? I am assuming that with $N=90$, the power of reliably identifying 5 clusters is quite low, but with a larger sample size, the findings may be different? This may be a great point to bring up in the discussion.

R34: We agree this is a possibility. As stated above, this possibility is now mentioned in the beginning of the "Limitations" section (page 41, lines 1239-1241), reading:

"[...] A larger scale replication of the paradigm would ensure greater power, for example, to identify more latent classes of behavior, providing a finer-grained picture of interindividual differences in social avoidance."

C35: p.35 – when comparing drift rate between Conditions and Class, is there a way to significantly test for an interaction? Or does the fact that the credible intervals for the condition differences overlap between Class A and Class B suggest that there is no interaction?

R35: We can test the interaction as the posterior distribution of difference of differences, i.e., $[(v_{\text{pred}} - v_{\text{unpred}}$ in class B) - $(v_{\text{pred}} - v_{\text{unpred}}$ in class A)]. A posterior distribution that does not include 0 in the 95%CrI would indicate that the interaction is credible, but this is not the case (95%CrI=[-.342, .400]). We are now reporting this result in the “*Exploratory analysis for behavioral heterogeneity – Results – Drift Diffusion Models*” section (page 34, lines 1019-1021).

C36: p.38 – how to interpret the finding that drift rate is much lower in Class A than Class B?

R36: We provided this interpretation in the Discussion, saying that this possibly relates to a greater sensitivity to threat in the SR vs. GD group, as supported by the greater EMG amplitude to threat at the moment of response (pages 37-38, lines 1131-1144). We now more clearly specify this interpretation, and the text reads as follows:

“[...] the SR Class showed a higher drift rate compared to the GD Class, with no difference between predictable and unpredictable conditions. This signaled that the value attributed to each option was unaffected by contingency degradation, speaking in favor of a SR process for value attribution. Furthermore, participants in the SR Class exhibited a slightly higher boundary separation and a greater drift rate regardless of the condition, and a longer non-decision time in the predictable condition than in the unpredictable condition, compared to the GD Class. Previous studies found that both the drift rate and the boundary separation increased in emotional context when the object of the decision is particularly salient or relevant¹⁰¹⁻¹⁰³, as well as the non-decision time¹⁰². Furthermore, in non-socioemotional perceptual decision-making, participants exhibited higher boundary separation and longer non-decision time¹⁰⁴ to improve accuracy by increasing response caution and/or adopting strategic motor slowing¹⁰⁵. Altogether, our results suggest that participants in the SR class may have been more sensitive to social threat, which diminished their ability to adapt to environmental contingencies in a goal-directed manner.”

C37: p.39 – I am not convinced that the results support a “direct causation of GD processes on social avoidance” – the manipulation of predictability may have caused an increase in social avoidance, but the causal role of GD processes cannot be known for sure, and may not be direct (i.e. other factors could be at play)

R37: We agree that, even in controlled experimental designs, potential confounding factors could always be present, and the sentence might have been too strong. We rephrased as this: “Overall, our results are consistent with the hypothesis of a central role for GD processes in social threat avoidance [...]” (page 36, lines 1082-1083).

C38: p.42 – if I understand the interpretation of Class A vs Class B correctly, it seems like people in Class A are more driven by GD processes, while those in Class B are more driven by SR processes. I understand this may be a bit simplistic and more nuance is needed; however given the grounding of the paper in the GD/SR dichotomy, it would be helpful to interpret the two classes in light of these processes (this relates to my major comment #1).

R38: We agree that interpreting the two classes in terms of GD and SR processes would make the manuscript clearer. We are now making this more explicit by: (a) we are now addressing the classes as “GD and SR class” instead of “Class A and B” and (b) we are now more clearly discussing this result in terms of the two processes.

Reviewer #3:

Thank you for giving me the opportunity to review the manuscript submitted to Communications Psychology by Sequestro et al. The paper reports on three experiments designed to investigate threat avoidance behaviour in a novel VR environment. Specifically, the aim of the study is to disentangle the contributions of stimulus-response associations and goal-directed processes to the avoidance of threats represented by angry avatar facial expressions. The main manipulation of the experimental design is full control over the avoidance of the threatening stimulus (predictable condition), contrasted with random confrontation with an angry avatar (.50), independent of initial avoidance decisions (unpredictable condition). While Experiments 1 and 2 focused on behavioural outcomes and their modelling, Experiment 3 additionally considered cardiovascular and muscular activity.

The main finding, consistent across the three experiments, is that avoidance rates are higher in predictable than unpredictable trials, with avoidance rates above chance level in the latter condition.

Although I am enthusiastic about the VR approach implemented in the study, as well as the authors' goal of replicating the main finding in the three experiments (all with proven sample size), I have a few major concerns about the study.

C39: In line with recent frameworks of approach-avoidance behavior, the authors propose two fundamentally different mechanisms underlying avoidance behavior, i.e., SR and GD processes. What I have missed in the introduction of the paper is a clear assumption as to whether SR and GD operate exclusively or – if not – run in parallel or serially. In my view, such a specification would be necessary to better motivate the rationale of the present study. What concerns me most, is that it remains unclear (from my careful reading of the paper) to what extent SR and/or GD processes are elicited by the different structures of predictable and unpredictable trials. Therefore, the two contrasting hypotheses derived at the end of the introduction seem very unspecific.

R39: Thank you for giving us the opportunity to clarify this point, which was raised by other reviewers, as well as the Editor. We realize that our previous formulation of the introduction was somehow misleading, as it suggested that the present work aimed at disentangling the serial vs. parallel nature of the SR vs. GD arbitration, which is not the point here. Therefore, we restructured entirely our Introduction, to better introduce our hypotheses. Namely, our line of reasoning proposes that:

- 1) Both SR and GD processes have been implicated in defensive responses to threat.
- 2) Contemporary models (e.g., Mobbs et al., 2009; 2020) suggest that these defensive modes are not dichotomous but lie on a continuum, which is influenced by the perceived distance to threat.
- 3) While it is clear that GD processes govern behavior under safety and pre-encounter phases, and that SR processes dominate under imminent threat, both are thought to intervene in post-encounter phases, when the threat has been detected but it is not attacking yet. Notably, this is relevant, as social threats in everyday life, such as when we encounter an angry or aggressive individual, most likely lie in this transition zone.

- 4) This raises the question on how the brain arbitrates between SR and GD defensive modes, when encountering aggressive individuals and when a fast approach/avoidance decision must be made.
- 5) It has been proposed that the controllability of expected action-outcomes governs the balance between SR and GD processes (Dorfman & Gershman, 2019). Notably, when the action-outcome contingency is degraded, and subjects have no control over the action outcomes, this should affect their behavior if it is GD, but not if it is SR.

This helped us to better formulate our hypotheses, to make clear predictions on how avoidance behavior should be affected, in case it is governed by SR processes, by GD ones, or by a combination of both. The hypotheses now read as this (page 5, lines 107-119):

“Overall, we expected that participants would avoid the angry avatar more than chance in the predictable condition as, irrespective of the type of process involved, both GD and SR processes should lead to threat avoidance. If GD processes are involved in the decision, then avoidance rates should decrease in the unpredictable context, because of contingency degradation. On the contrary, such reduction should not be observed if avoidance is solely driven by SR associations. Importantly, a simple reduction in avoidance rate under unpredictability would not imply that SR processes are not involved in the decision, as SR processes might just keep operating in the unpredictable context as they do in the predictable one. Conversely, an avoidance rate at chance level in this condition would suggest that SR associations are not driving avoidance, either because SR action tendencies in response to threatening expressions are simply not elicited, or because they are not overtly influencing avoidance. Therefore, by comparing the avoidance patterns between the two conditions, we tested the effect of GD and SR processes over social-threat avoidance.”

C40: In general, the study lacks specific hypotheses, especially given the large number of variables collected, including individual trait parameters, RTs included in the avoidance rate models, and physiological parameters. Regarding the latter, for example, there is no scientific justification for measuring EMG activity of the flexor pollicis brevis muscle in addition to the overt responses required by the task decisions (there is only a single sentence on p. 21, which is not clearly linked to the aims of the paper). Trait variables (obtained from questionnaires) are not included in the models of the data from Experiments 1 and 2, but are then implemented in Experiment 3, again without any directed hypotheses. Similarly, the inclusion of the Subjective Evaluation Task in Experiment 3 appears to be incidental, and it remains unclear why the authors related its scores to some of the variables of interest but not to others. In summary, the paper seems to report a series of exploratory analyses which, unfortunately, are not convincingly motivated.

R40: We acknowledge that the former formulation of the introduction and our hypotheses might have induced confusion on our scientific approach over the three experiments.

We hope that now our main hypotheses are clearer. Experiments 1 and 2 focused on behavior, which alone sufficed to respond to our main question, notably the relative contribution of SR and GD processes in social threat avoidance decisions. The results of these first experiments suggested a strong

implication of GD processes in such decisions. In light of this, Experiment 3 aimed on one hand at replicating behavioral findings, and on the other hand at clarifying them in the context of recent embodied frameworks, suggesting an important role of physiological responses in the integration of subjective value in the decision (Hashemi et al., 2019; Livermore et al., 2021; Skora et al., 2022). We disagree with the Reviewer that this is not well motivated in the paper, as the introduction of Experiment three (pages 19-20, lines 572-589) details these models, providing clear justification for both the cardiac and the subjective measures. Specifically, the results showing that only in trials in which a strong cardiac deceleration in response to the stimulus is observed participants integrate the subjective value of the predicted outcome (approach/avoidance) in their decision are of relevance, as they support these newly proposed models, and better clarify the embodied nature of GD processes guiding decisions in the face of emotional expressions.

On the other hand, we agree that the EMG measure, as well as the questionnaires, were used as exploratory measures, and were not sufficiently motivated. We now provided more thorough justification in the text. Concerning EMG, we added (page 20, lines 590-599):

“Finally, as an exploratory analysis, we measured EMG activity of response effector muscles during the main task. Muscle activity, such as involuntary hand muscles contraction following transcranial magnetic stimulation of the primary motor cortex, measured with EMG^{77,78}, or sustained force exerted due to voluntary isometric wrist extension⁷⁹, has been used as an objective marker of action readiness during viewing of emotional images. Similarly, we measured muscle contraction of the flexor pollicis brevis with EMG, at the moment of response production, as a measure of action readiness. Here, we wanted to explore whether action readiness is modulated either by the simple presence of threat in the scene, or by the predictability of the outcome, which would clarify whether both SR and GD mechanisms can influence motor preparation.”

In the Discussion we interpret our findings by speculating that greater EMG amplitude in response to threat scenes, only in the SR class, might be a peripheral correlate of an increased SR responsiveness to threatening information (pages 38-39, 1166-1177). Clearly, future studies are needed to specifically address this hypothesis in dedicated experiments.

Concerning questionnaires, we already responded to Reviewer 2 who expressed a similar concern, so please see R10.

C41: Having gained this impression, I was rather surprised by the final "exploratory analysis" section of the paper. The authors justify these additional analyses with the assumption of potential heterogeneity in avoidance behaviour. I fully agree with the authors that such inter-individual differences are usually ignored in experimental studies and that the cumulative power from the three independent samples of their experiments is sufficient to take them into account. However, it remains unclear on what the authors base this assumption in their study; this would be more comprehensible if they had compared models for the individual experiments, e.g. with versus without 'subjectID' as a predictor.

R41: Thank you for the opportunity to provide a better clarification of this analysis. The point here was to test whether the co-presence of behavioral signatures of SR and GD processes throughout the three experiments (i.e., avoidance decreases after contingency degradation in the unpredictable condition – GD – but remains above chance – SR) was due to intra-individual variability in the behavioral strategy used (i.e., subjects would use a mixture of SR and GD strategies), or to inter-individual variability. We now extensively motivate this analysis (pages 30-31, lines 919-941) as follows:

“Across three experiments we found that participants avoided virtual angry avatars more frequently when they could predict the outcome of their decisions, but still more than chance when such prediction was not possible due to action-outcome contingency degradation (i.e., in the unpredictable condition). While a possible interpretation of this pattern entails intra-individual variability in the use of GD and SR strategies (e.g., using both in competition and in both conditions, but only SR in the unpredictable context), an alternative explanation entails inter-individual variability, meaning the presence of several latent behavioral patterns⁹⁰⁻⁹³. Indeed, a limitation of common statistical models is that they assume homogeneity of behavior, with inter-individual differences typically interpreted as variance around a central tendency. Nonetheless, heterogeneity in free choice tasks^{34,94}, and especially in avoidance behavior, is the norm in both animal and human populations^{90-92,95}. This seems relevant in the current study's case and especially in the interpretation of our behavioral results. Specifically, it is possible that some participants avoided above chance in both conditions and with no difference between them, irrespective of contingency degradation, in an SR fashion. Conversely, other participants may have learned from outcomes and avoided at chance level in the unpredictable condition and significantly more in the predictable condition, in a GD fashion. Using classical statistical modeling and aggregating all participants to make inferences about single parameter estimates would fail in capturing such inter-individual variability, which may explain the observed behavioral pattern. To address this possibility and explore possible latent classes in our sample, we used Finite Mixture Modeling. Mixture models assume that data are sampled from a mixture of generating processes, or different populations. This method allows for data-driven clustering within datasets without any a-priori knowledge.”

Concerning whether taking as random effect the SubjectID ameliorated model fit, this is evident from the random effect tables for each model for each experiment presented in the Supplementary Material (*Tables D3, D10, D17, D24*). These results show a substantial variability across subjects in each parameter of the model.

C42: Taken together, the paper reports interesting - albeit predictable - findings from a novel VR paradigm. However, in my view, the overarching research question regarding the contribution and weighting of GD and SR processes in avoidance decisions remains unanswered, which may be due to the operationalization. The manuscript is rich in data and analyses but simultaneously lacks specific hypotheses. As a result, the general discussion (and conclusion section) is inconclusive and often repetitive.

R42: We have now substantially changed the Introduction, as well as the Discussion following Reviewer's suggestions and responding to criticisms, and we are confident that hypotheses are now clearer, as well as the contribution of the present results.

Response to reviewers

Manuscript Number: COMMSPSYCHOL-24-0069B

“Social threat avoidance depends on action-outcome predictability”

Reviewer #1:

C1: The authors have appropriately addressed my comments - I am now happy to recommend the manuscript for publication.

R1: We thank the reviewer for their positive evaluation of our reviewed manuscript.